# TMX4-driven LINC complex disassembly and asymmetric autophagy of the nuclear envelope upon acute ER stress

Marika K. Kucińska[1,2,7], Juliette Fedry [3,7], Carmela Galli[1], Diego Morone [1,4], Andrea Raimondi[1,5], Tatiana Soldà[1], Friedrich Förster [3] & Maurizio Molinari [1,6] ✉

The endoplasmic reticulum (ER) is an organelle of nucleated cells that produces proteins, lipids and oligosaccharides. ER volume and activity are increased upon induction of unfolded protein responses (UPR) and are reduced upon activation of ER-phagy programs. A specialized domain of the ER, the nuclear envelope (NE), protects the cell genome with two juxtaposed lipid bilayers, the inner and outer nuclear membranes (INM and ONM) separated by the perinuclear space (PNS). Here we report that expansion of the mammalian ER upon homeostatic perturbations results in TMX4 reductase-driven disassembly of the LINC complexes connecting INM and ONM and in ONM swelling. The physiologic distance between ONM and INM is restored, upon resolution of the ER stress, by asymmetric autophagy of the NE, which involves the LC3 lipidation machinery, the autophagy receptor SEC62 and the direct capture of ONM-derived vesicles by degradative LAMP1/RAB7-positive endolysosomes in a catabolic pathway mechanistically defined as *micro*-ONM-phagy.

The first indication of a membrane system surrounding the nucleus was published in 1833 by Robert Brown[1]. The high resolution offered by electron microscopy subsequently revealed that this membrane system, the nuclear envelope (NE), consists of two lipid bilayers, the outer nuclear membrane (ONM) and the inner nuclear membrane (INM), which are a continuation of the ER membrane and delimit the perinuclear space (PNS), which is contiguous to the ER lumen (Fig. 1a). These pioneering studies performed first in amphibian oocyte nuclei[2,3] and subsequently in mammalian cells[4,5] determined the even distance of about 20–23 nm between the two lipid bilayers and reported on the presence in the NE of nuclear pore complexes with a diameter of about 100 nm[2,3,6].

Assessment of mechanisms that regulate NE dynamics is relevant because cumulating data reveal that the NE, a subcellular structure that distinguishes eukaryotic from prokaryotic cells, is not a simple barrier that confines nucleoplasmic activities such as gene regulation and transcription from cytoplasmic activities such as gene translation into polypeptide chains. Rather, the NE is a very dynamic structure[7,8], which is deconstructed and then re-assembled from ER membranes during open mitosis in metazoan[9], controls access of macromolecules to the nucleoplasm via nuclear pore complexes[10], forms double-membrane projections in the cytoplasm as observed in several tumors and laminopathies, or invaginations[11,12], it is remodeled to respond to mechanical forces[13].

Linker of Nucleoskeleton and Cytoskeleton (LINC) complexes play a crucial role in the control of NE dynamics. They are formed by SUN proteins in the INM that are covalently linked via *inter*molecular disulfide bonds with NESPRIN proteins in the ONM (Fig. 1a)[14,15]. These complexes connect the nucleoskeleton with the cytoskeleton, control nuclear positioning and sense mechanical forces[14,16–19]. LINC complexes

[1]Università della Svizzera italiana (USI), Faculty of Biomedical Sciences, Institute for Research in Biomedicine, CH-6500 Bellinzona, Switzerland. [2]Department of Biology, Swiss Federal Institute of Technology, CH-8093 Zurich, Switzerland. [3]Structural Biochemistry, Bijvoet Center for Biomolecular Research, Utrecht University, 3584 CG Utrecht, The Netherlands. [4]Graduate School for Cellular and Biomedical Sciences, University of Bern, CH-3000 Bern, Switzerland. [5]Experimental Imaging Center, San Raffaele Scientific Institute, I-20132 Milan, Italy. [6]School of Life Sciences, École Polytechnique Fédérale de Lausanne, CH-1015 Lausanne, Switzerland. [7]These authors contributed equally: Marika K. Kucińska, Juliette Fedry. ✉e-mail: maurizio.molinari@irb.usi.ch

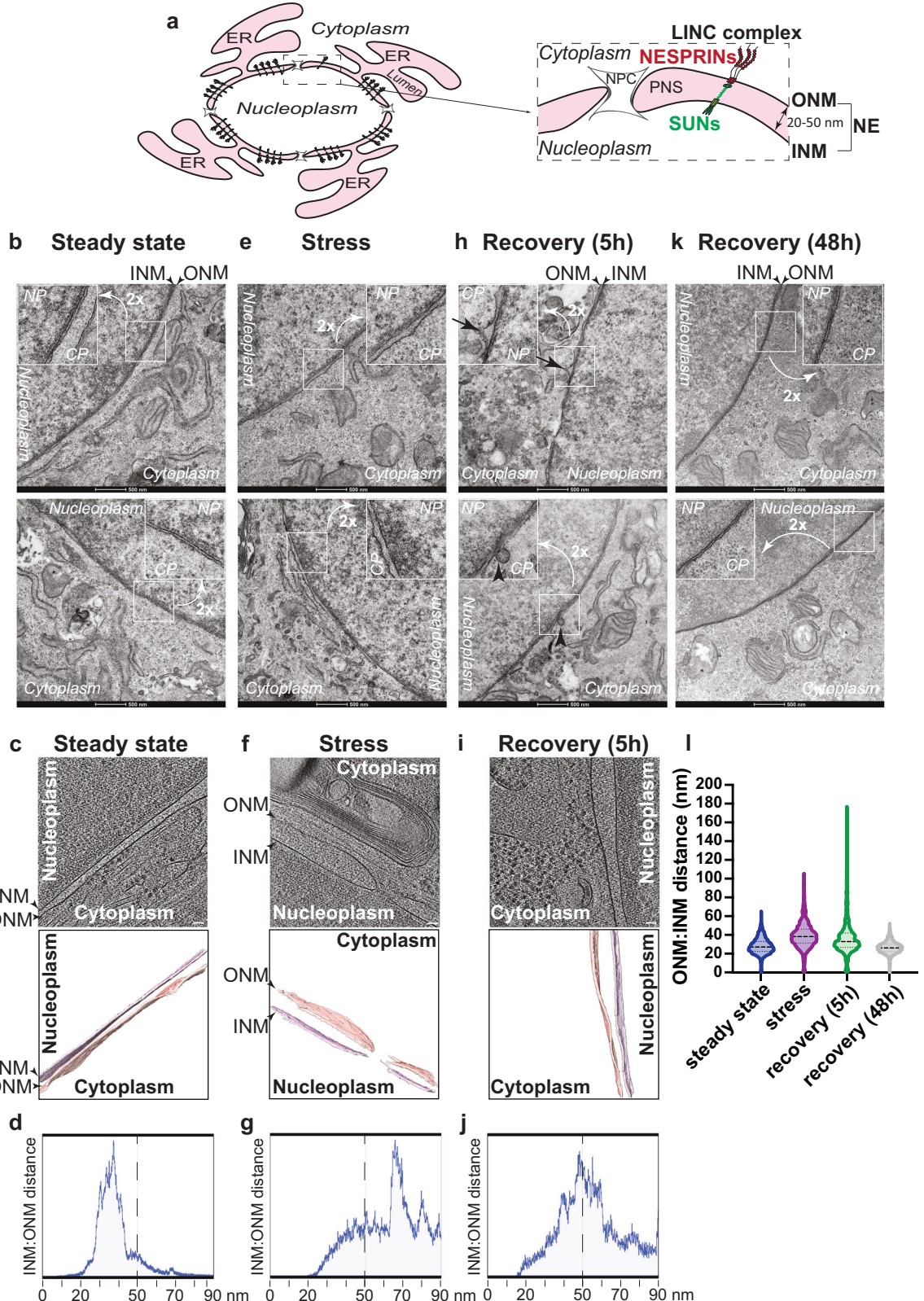

contribute to maintain the width of the PNS below 50 nm[14,20,21] at least in adherent cells and in cells characterized by increased mechanical tension[22,23].

Cells can adapt the size of the ER to their needs. Physiologic and pathologic conditions such as cell activation, pharmacologic treatments, accumulation of misfolded proteins or pathogen invasion activate a group of stress sensors in the ER membrane (IRE1, ATF6,

PERK). These trigger anabolic unfolded protein responses (UPR) resulting in transcriptional and translational induction of a subset of genes eventually leading to the expansion of the ER volume and activity[24]. Interruption of the stress signal and/or resolution of the stress condition activates autophagic programs named recov-ER-phagy that resume physiologic ER size and activity upon ER fragmentation and lysosomal clearance of excess ER portions[25]. Recov-ER-

**Fig. 1 | Morphometric changes of the NE upon ER stress and during recovery from ER stress. a** Schematic representation of the ER and the NE. The inset shows the LINC complex that spans the perinuclear space (PNS) between the INM and the ONM. NPC is a nuclear pore complex. **b** RT-TEM micrograph showing two representative MEF at steady state. Nucleoplasm (NP in the inset), cytoplasm (CP) and the double lipid bilayer (ONM and INM) constituting the NE are shown. **c** Upper panel showing a slice through a CET of the NE from a lamella through a MEF at steady state. Scale bar: 50 nm. The lower panel represents the isosurface representation of the corresponding segmented volumes for NE at steady state. ONM in salmon; INM in light purple. **d** Quantifications of corresponding ONM-INM distances at steady state (see Tomogram 14, Supplementary Fig. 1a). **e** Same as (**b**) during the perturbation of ER homeostasis with CPA. **f** Same as (**c**) under ER stress. Scale bar: 50 nm. **g** Same as (**d**) for a cell under ER stress (see Tomogram 1, Supplementary Fig. 1b). **h** Same as (**b**) in MEF recovering from ER stress (5 h after interruption of CPA exposure). **i** Same as (**c**), for a recovering MEF. Scale bar: 50 nm. Please also refer to Movie 1. **j** Same as (**d**) for a recovering MEF (see Tomogram 1, Supplementary Fig. 1c). **k** Same as (**b**) 48 h after interruption of the pharmacologic treatment. **l** Violin plot showing INM:ONM distances in five representative MEF examined by RT-TEM for each condition. The dotted line shows the average width of the PNS.

phagy relies on activation of the ER-phagy receptor SEC62 that, during recovery from acute ER stresses, associates with the cytosolic ubiquitin-like protein LC3 to drive piecemeal *micro*-ER-phagy programs that involve the LC3 lipidation machinery and ESCRT-driven capture of ER-derived vesicles by LAMP1/RAB7-positive endolysosomes for clearance[26–28].

Thus, the ER is a plastic organelle, whose size and activity are modulated by anabolic and catabolic programs[25]. How and if the enlargement or the collapse of the ER volume is transmitted to the NE is poorly understood. Studies in the yeast *Saccharomyces cerevisiae* initially showed that homeostatic perturbations that trigger UPR and dramatically enlarge the ER lumen are not transmitted to the NE, which resists membrane and lumen expansion despite the physical continuity with the ER[29–31]. More recently, however, expansion of the yeast PNS has been reported in response to perturbation of protein folding[32]. In mammalian cells, the ONM can expand by the addition of lipids from the ER[33] and the INM and ONM must diverge, and the PNS must substantially be expanded, to allow nuclear egress of herpes virus capsids[34,35], or ribonucleoprotein particles[36]. The physiologic and pathologic regulation of NE dynamics is poorly understood.

Here, by monitoring mammalian cultured cells exposed to pharmacologic perturbation of ER homeostasis, we show that expansion of the mammalian ER is transmitted to the NE upon disassembly of LINC complexes and results in swelling of the ONM and enlargement of the PNS. Physiologic NE ultrastructure is restored upon termination of the pharmacologic treatment, which activates selective autophagy of excess ONM subdomains. We report on the role of the ONM-resident TMX4 reductase in the disassembly of the LINC complexes upon NE swelling and of LC3 lipidation-dependent, SEC62-driven capture and clearance of excess ONM portions by RAB7/LAMP1-positive endolysosomes to restore the physiologic NE ultrastructure on the resolution of the ER stress condition.

## Results

### Morphometric analyses reveal expansion of the mammalian NE upon CPA-induced stress

The ultrastructure and the dynamic changes of the double lipid bilayer constituting the NE of mouse embryonic fibroblasts (MEF) upon perturbation of ER homeostasis were examined by room temperature-transmission electron microscopy (RT-TEM) of chemically fixed, resin-embedded cells and by cryogenic Focus Ion Beam milling followed by cryo-electron tomography (FIB-CET)[37] of vitrified cells. At steady state, RT-TEM (Fig. 1b and Insets, 2 cells are shown) and FIB-CET micrographs (Fig. 1c) reveal that the INM and the ONM are generally evenly spaced, as previously reported[38,39]. The distance between INM and ONM at steady state generally remains below the 50 nm allowed by the maximal extension of the LINC complexes[14,20–23] (Fig. 1d, Supplementary Fig. 1a). To assess whether the ER swelling elicited by a pharmacologic treatment that perturbs ER homeostasis is transmitted to the NE, we exposed MEF for 12 h to 10 μM cyclopiazonic acid (CPA), a reversible inhibitor of the sarco/ER calcium pump[26,28,40]. Cell exposure to CPA mimics ER stresses triggered by perturbation of glycosylation (tunicamycin[41,42]), redox (DTT[26,42,43]), or calcium homeostasis (thapsigargin[42,44]), which are characterized by ER expansion and induction of ER-resident

proteins[26,28,40]. Analyses of the RT-TEM micrographs (Fig. 1e and Insets, 2 cells are shown) and of FIB-CET micrographs (Fig. 1f ) reveal that the PNS is enlarged in MEF exposed to CPA and that large subdomains of the NE shown by distances between INM and ONM well above the 50 nm mark (Fig. 1g, Supplementary Fig. 1b). The average thickness of the NE at steady state as determined upon manual segmentation in Microscopy Image Browser (MIB)[45] is about 27 nm (Fig. 1l, Steady state, NE thickness of 5 individual cells), a value that approximates well the thickness established in pioneering work published in the 1950s and confirmed in more recent studies[2–6,21,38,39]. The violin plots show that only in a few discrete NE subdomains, the PNS width may exceed 50 nm (Fig. 1l, for RT-TEM analyses and Supplementary Fig. 1d for the probability to exceed the 50 nm width as determined by FIB-CET analyses at Steady state). During the pharmacologic treatment with CPA, the average NE thickness raises by 40% to a value of about 38 nm (Fig. 1l, Stress), with portions of the ONM that may swell to increase the width of the PNS up to 120 nm (Fig. 1l and Supplementary Fig. 1d for the probability to exceed the 50 nm width as determined in cells exposed to ER stress).

### Asymmetric NE vesiculation and return to physiologic NE ultrastructure upon recovery of ER homeostasis

ER swelling upon UPR induction and subsequent activation of catabolic programs that restore the physiologic ER size upon interruption of pharmacologic treatments have originally been documented for hepatocytes in mice subjected to acute (i.e., transient) treatments with the anti-epileptic drug phenobarbital[46–48]. The mechanisms reducing the size of the ER during recovery of the physiologic ER homeostasis have been eventually recapitulated and characterized upon acute treatments of cultured cells with CPA[26,28]. These studies from our lab show that *recov*-ER-phagy programs that restore physiologic ER size and content rely on vesiculation of ER subdomains and their capture by RAB7/LAMP1-positive endolysosomes for clearance via piecemeal *micro*-ER-phagy[26–28]. We reasoned that these recovery programs could also reshape the NE to its physiologic status upon the termination of acute ER stresses that cause deformation of the ONM.

RT-TEM and FIB-CET micrographs reveal that 5 h after interruption of the pharmacologic treatment with CPA, portions of the ONM project into the cytoplasm (the distance between INM and ONM may locally raise above 180 nm, Fig. 1h, Insets, arrows in the upper panel, Fig. 1i, j, l). Vesicles originating from the ONM can also be seen (Fig. 1h, lower panel, arrowhead, Movie 1 and results shown below).

Room temperature-electron tomography (RT-ET) of subdomains of the NE may reveal, as in the example shown in Fig. 2a, the initial phase of ONM budding (Fig. 2a, arrowhead 1 and Movie 2), the closure of the bud that precedes detachment of the ONM-derived vesicle (Fig. 2a, arrowhead 2 and Movie 2), and the detachment of an ONM-derived vesicle simultaneously proceeding in a small area (about 0.04 μm²) of the NE (Fig. 2a, arrowheads 3a and 3b, Movie 2).

Swelling and vesiculation of the ONM gradually decrease with the prolongation of the recovery phase. Forty-eight hours after interruption of the pharmacologic treatment, the ultrastructure of the NE (Fig. 1k, l) is morphologically undistinguishable from the NE ultrastructure in untreated cells (Fig. 1b). These results show that like the

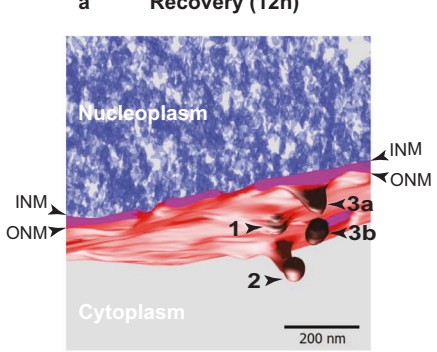

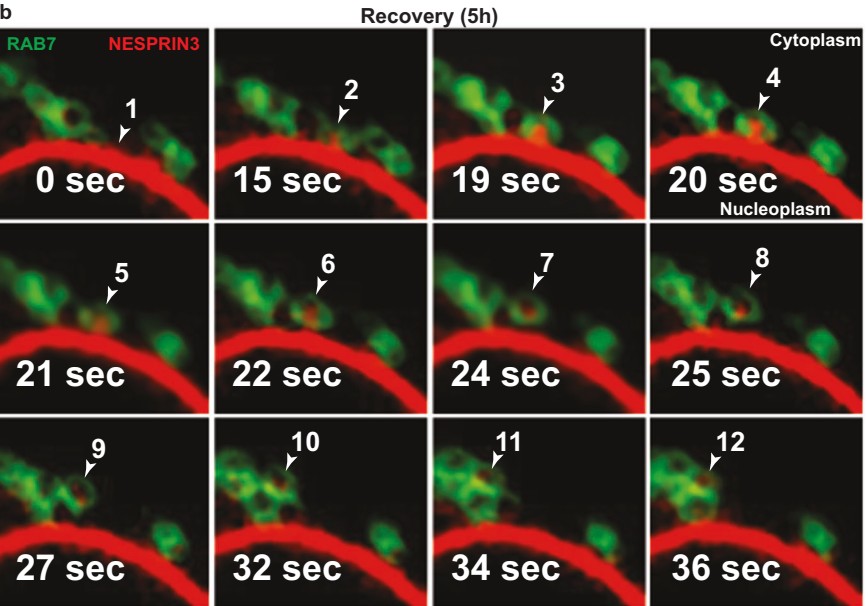

**Fig. 2 | Asymmetric vesiculation of the NE and capture by endolysosomes.**
**a** Room temperature-electron tomography of a subdomain of the NE in a MEF recovering from ER stress (12 h after interruption of CPA exposure). INM and ONM are shown with arrowheads. (1) The initial stage of ONM deformation, (2) the formation of an ONM-derived vesicle, (3) a vesicle has just been released from the ONM. Refer to Movie 2. **b** Selected frames of Movie 3 showing the capture of a HALO-NESPRIN3-positive ONM-derived vesicle (arrowheads) by GFP-RAB7-positive endolysosomes during recovery from ER stress.

ER[26–28], the NE returns at pre-stress morphology after resolution of homeostatic perturbations.

## Capture of ONM portions by endolysosomes during recovery from ER stress

To better characterize NE dynamics and to monitor the fate of ONM-derived vesicles generated during the recovery phase initiated upon withdrawal of CPA, we labeled the ONM with the resident protein NESPRIN3α[49,50] fused with HaloTag, a bacterial hydrolase, whose active site has been modified to facilitate covalent and irreversible binding of cell-permeable fluorescent ligands (Supplementary Fig. 2a)[51]. Time-course analyses by confocal light scanning microscopy (CLSM) of living cells imaged 5 h after interruption of the CPA treatment (Fig. 2b and Movie 3) reveal the detachment of HALO-NESPRIN3α-positive portions from the NE and their capture by degradative GFP-RAB7-positive endolysosomes (Movie 3 and selected frames in Fig. 2b).

The generation of NESPRIN3α-positive ONM portions and their capture by endolysosomes is monitored in cells at steady state (Fig. 3a, upper panels) or recovering from ER stress (Fig. 3a, lower panels). Cell exposure to BafilomycinA1 (BafA1) inhibits hydrolytic lysosomal activity and reveals the substantial increase of HALO-

NESPRIN3α-positive ONM portions within LAMP1-positive endolysosomes in cells recovering from ER stress (12 h after CPA withdrawal, Fig. 3a, lower panels). The increase of lysosomal delivery of ONM portions during recovery from ER stress is confirmed by quantifications with LysoQuant (Fig. 3b), an unbiased and automated deep learning image analysis tool for segmentation and classification of fluorescence images capturing cargo delivery to lysosomal compartments (ref. 52 and freely available at www.irb.usi.ch/lysoquant/). GFP-tagged SUN1 (Fig. 3c, d) and endogenous SUN2 (Supplementary Fig. 2b, c), two protein markers of the INM[15,53], are not delivered within endolysosomes under the same experimental conditions. This confirms the selective vesiculation and lysosomal delivery of ONM portions during recovery from ER stress (Fig. 1h, lower panel, 2a, 2b and results below).

## The LC3 lipidation machinery controls capture of ONM portions by LAMP1-positive endolysosomes

Vesiculation and lysosomal delivery of organelle portions for clearance involve autophagy gene products regulating lipidation of the ubiquitin-like protein LC3, and organelle-specific LC3-binding autophagy receptors displayed at the limiting membrane of the organelle

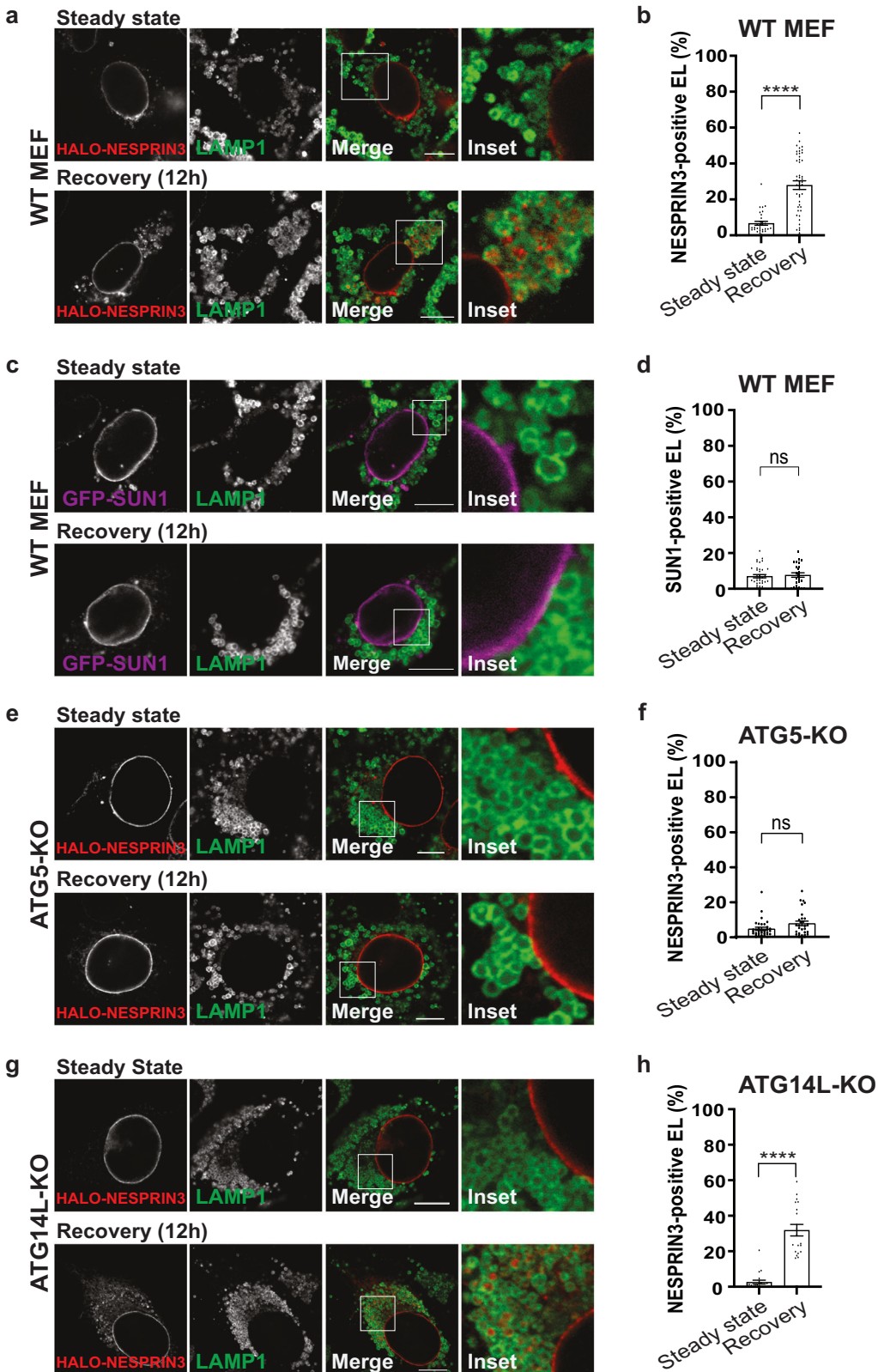

portion to be removed from cells[25,54,55]. Inspired by these findings, we first assessed lysosomal delivery of NESPRIN3-positive ONM portions in cells lacking ATG5, i.e., defective in LC3 lipidation[56]. As reported for other types of organello-phagy[25,54,55], LysoQuant analyses reveal that the absence of ATG5 substantially inhibits the delivery of NESPRIN3-positive ONM portions to the lysosomal compartments during the recovery phase from ER stress (Fig. 3e, f). In contrast, delivery of

NESPRIN3-positive ONM portions to the LAMP1-positive degradative organelles proceeds normally in cells lacking ATG14L (Fig. 3g, h), which is dispensable for LC3 lipidation but is involved in autophagosome formation[57–60] and in control of membrane tethering and autophagosome fusion with lysosomes[61].

The dispensability of autophagosome biogenesis and LC3 lipidation-dependent capture of ONM subdomains by degradative

**Fig. 3 | Lysosomal delivery of ONM subdomains. a** CLSM analyses of HALO-NESPRIN3α-positive ONM subdomains delivery within LAMP1-positive endolysosomes in WT MEF at steady state (upper panels) or in MEF recovering from ER stress, 12 h after interruption of the pharmacologic treatment with CPA (lower panels). Cells incubation with BafA1 prevents clearance of cargo eventually delivered within endolysosomes. Scale bars: 10 μm. **b** Quantification of (**a**) by LysoQuant[52]. n = 32 and 47 cells for steady state and recovery, respectively. N = 3 independent experiments. Mean ± SEM; unpaired, two-tailed t-test, ****P < 0.0001. **c** Same as (**a**), to monitor lysosomal delivery of GFP-SUN1. Refer to Supplementary

Fig. 2b, c for lysosomal delivery of endogenous SUN2. **d** Quantification of (**c**). n = 37 and 25 cells for steady state and recovery, respectively. N = 3 independent experiments, mean ± SEM; unpaired, two-tailed t-test, ns. P > 0.05. **e** Same as (**a**) in MEF lacking the autophagy gene product ATG5[56]. **f** Quantification of (**e**). n = 33 and 29 cells for steady state and recovery, respectively. N = 2 independent experiments, mean ± SEM; unpaired, two-tailed t-test, ns. P > 0.05. **g** Same as (**a**) in MEF lacking the autophagy gene product ATG14L[57-60]. **h** Quantification of (**g**). n = 21 and 19 cells for steady state and recovery, respectively. N = 2 independent experiments, mean ± SEM; unpaired, two-tailed t-test, ***P << 0.0001.

LAMP1/RAB7-positive endolysosomes (Figs. 2b and 3e–h and results below), match the mechanisms that restore physiologic ER volume and activity after resolution of ER stresses, which proceeds via *micro-autophagy*[26–28].

### The autophagy receptor SEC62 regulates delivery of ONM portions to LAMP1-positive endolysosomes

Recovery from acute ER stresses activates the function of SEC62 as an autophagy receptor[26,28]. Relevantly, immunogold EM shows that endogenous SEC62 distributes in the ER (arrowheads in Fig. 4a) and in the ONM (arrows in Fig. 4a and Inset). Immunogold EM micrographs of MEF during the recovery phase, reveal the presence of endogenous SEC62 at the limiting membrane of bulges protruding from the ONM (arrow in Fig. 4b and inset), at the limiting membrane of vesicles released from the ONM in proximity to endolysosomes (arrow 1 in Fig. 4c and inset), and at the site of contact between the ONM and endolysosomes (arrow in Fig. 4d and inset). Under the same experimental conditions, immunogold EM shows the localization of HALO-NESPRIN3α in the ONM (arrow in Fig. 4e and inset) and at the limiting membrane of vesicles in the endolysosomal lumen (arrowheads in Fig. 4e and inset).

To assess the possible involvement of SEC62 as the autophagy receptor regulating lysosomal clearance of the NESPRIN3-positive ONM portions, we made again use of HALO-NESPRIN3α as protein marker of the ONM and compared its delivery to the LAMP1-positive endolysosomal compartment in WT MEF and in MEF lacking SEC62 generated by CRISPR-Cas9 genome editing (Supplementary Fig. 3a, lanes 2, 5–7)[26,28]. The CLSM and LysoQuant analyses confirm the increased delivery of HALO-NESPRIN3α to the LAMP1-positive degradative compartment in CRISPRWT MEF recovering from ER stress (Fig. 5a, b, Steady state vs. Recovery). In cells lacking the autophagy receptor SEC62 (CRISPRSEC62), HALO-NESPRIN3α delivery to the lysosomal compartment during the recovery phase is substantially inhibited (Fig. 5c, d, Steady state vs. Recovery). The back-transfection of SEC62 resumes delivery of HALO-NESPRIN3α to the LAMP1-positive endolysosomes (Fig. 5d, +SEC62, Fig. 5e, Supplementary Fig. 3a, lane 6). The back-transfection of a variant of SEC62, where the LC3-binding activity has been abolished upon mutation of the phenylalanine-glutamic acid-methionine-isoleucine LIR sequence to tetra-alanine (as in refs. 26,28), does not resume delivery of HALO-NESPRIN3α within the LAMP1 compartment (Fig. 5d, +SEC62LIR, Fig. 5f, Supplementary Fig. 3a, lane 7). Deletion of the starvation-induced ER-phagy receptor FAM134B[62] does not affect delivery of HALO-NESPRIN3α to the endolysosomal compartment during recovery from ER stress showing dispensability of this autophagy receptor for lysosomal turnover of ONM subdomains (Supplementary Fig. 3b–d). Figure 5g shows the localization of HALO-NESPRIN3α in the NE and the distribution of endogenous SEC62 in the ER and NE at steady state. During recovery from ER stress, both HALO-NESPRIN3α, endogenous SEC62 (Fig. 5h) and endogenous LC3 (Supplementary Fig. 2d) accumulate within inactive LAMP1-positive endolysosomes. This is consistent with the capture by endolysosomes of SEC62-positive ER-derived vesicles (as also shown in refs. 26,28 and Fig. 4c) and of SEC62/HALO-NESPRIN3α-positive ONM-derived vesicles.

### LINC complexes are disassembled for transmission of the ER enlargement to the NE upon ER stress

Our data show that the deformation of the ONM upon perturbation of ER homeostasis increases the width of the mammalian PNS above the 50 nm determined by the LINC complexes[14,20–23] in adherent cells and in cells characterized by increased mechanical tension[22,23]. To visualize the PNS in vitrified MEF cells, we selected the thinnest FIB-CET tomogram in each condition, giving the highest contrast in the NE region. We deconvoluted the tomogram and masked stronger densities to guide the manual segmentation of densities connecting the ONM and INM. The analyses of cells at steady state (Fig. 6a, b, Supplementary Fig. 4a, b, Movie 4), upon exposure to ER stress (Fig. 6c, d, Supplementary Fig. 4c, d, Movie 5), or during recovery from ER stress (Fig. 6e, f, Supplementary Fig. 4e, f, Movie 6) indicate the presence of continuous densities bridging the INM and ONM only in NE subdomains, where the distance between the lipid bilayers is below the 50 nm (Fig. 6b, d, f and arrowheads in Supplementary Fig. 4). Above this limit, we did not observe filaments connecting the INM and the ONM (Fig. 6b, d, f, Supplementary Fig. 4a–f, Movies 4–6). These results are in good agreement with structural data on the elastic perinuclear domains of SUN trimers, which can maximally elongate up to 45–50 nm (Fig. 1a)[14,20–23] and with a report showing that LINC complexes maintain the width of the PNS within this limit[22,23].

### Endogenous NESPRIN proteins are clients of the reductase TMX4 in cellula

The results shown so far led us to explore the possibility that conditions of cellular stress activate an enzyme-driven pathway that remodels the NE. This would promote/allow a substantial increase of the distance between ONM and INM upon reduction of the *inter*molecular disulfide bond linking the perinuclear domains of SUN and NESPRIN proteins, thus disassembling the LINC complexes. In our model (Fig. 6g), the dissociation of SUN proteins from NESPRIN proteins would allow the increase of the width between the two membranes to the values measured in cells exposed to the pharmacologic treatment (Fig. 1l, Supplementary Fig. 1).

More than 20 oxidoreductases of the protein disulfide isomerase (PDI) superfamily populate the ER[63]. The intracellular distribution of a few of them, i.e., endogenous PDI, ERp57, ERp72 and ectopically expressed TMX3, TMX4, TMX5 are shown as examples (Fig. 7a–f, respectively). The type I ER membrane proteins TMX3, TMX4 and TMX5 also decorate the NE, which is continuous with the ER membrane (Fig. 7d–f and insets).

The biological functions of TMX3, TMX4 and TMX5 remain to be established[64]. However, some information is available in the literature. TMX3 is characterized by a canonical cysteine-glycine-histidine-cysteine (CGHC) active site sequence and acts as an oxidase in vitro[65]. TMX4 has a non-canonical cysteine-proline-serine-cysteine (CPSC) active site sequence. The proline residue at position 2 destabilizes the disulfide state and favors the di-thiol reduced form of the active site[66]. Consistently, TMX4 acts as reductase in vitro[67] and is therefore a good candidate as the enzyme involved in LINC complexes disassembly upon reduction of the *inter*molecular disulfide bonds that covalently link NESPRIN and SUN proteins. Moreover, TMX4 has a peculiar enrichment in the NE[68] (Fig. 7e and Inset). Previous analyses by

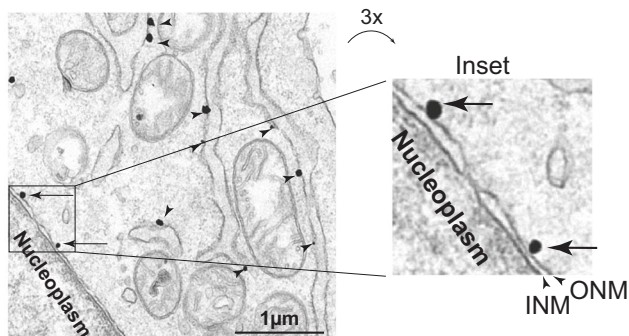

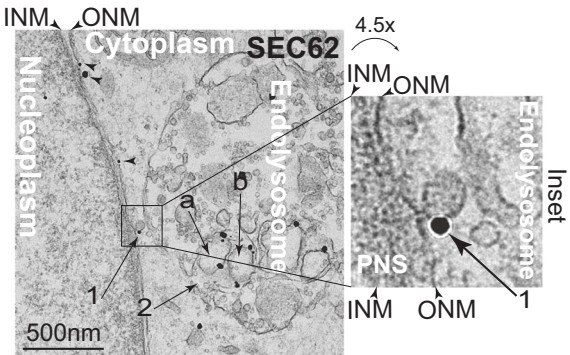

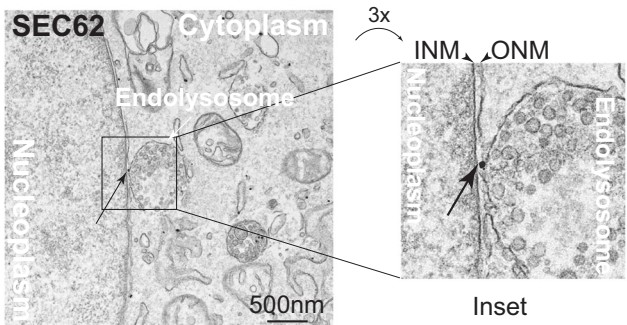

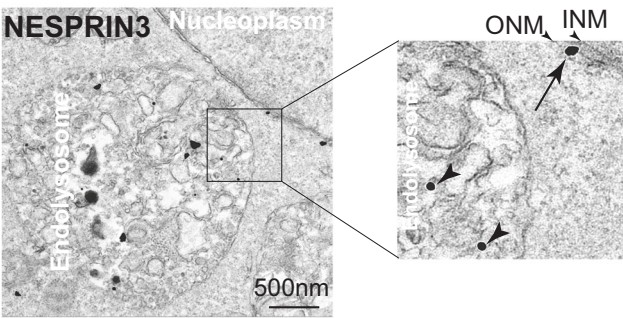

**Fig. 4 | Subcellular distribution of endogenous SEC62. a** RT-TEM micrograph showing immunogold labeling of endogenous SEC62 in the ER (arrowheads) and in the ONM (arrows) of MEF at steady state. **b** Same as (**a**) in MEF recovering from ER stress (12 h after interruption of the CPA treatment and exposure to 50 nM BafA1). Arrow and Inset show SEC62 in a bulge formed by the ONM. **c** Same as (**b**), where SEC62 (arrow 1) labels the limiting membrane of a vesicle caught in the act of detaching from the ONM. An endolysosome capturing SEC62-positive vesicles (**a** and **b**) by *micro*-autophagy is shown. Arrow 2 shows the site of vesicle engulfment. **d** Same as (**b**), where endogenous SEC62 is in a site of contact between the ONM and the endolysosome. **e** Same as (**b**), for HALO-NESPRIN3α at the ONM (arrow) and within an endolysosome next to the NE (arrowheads).

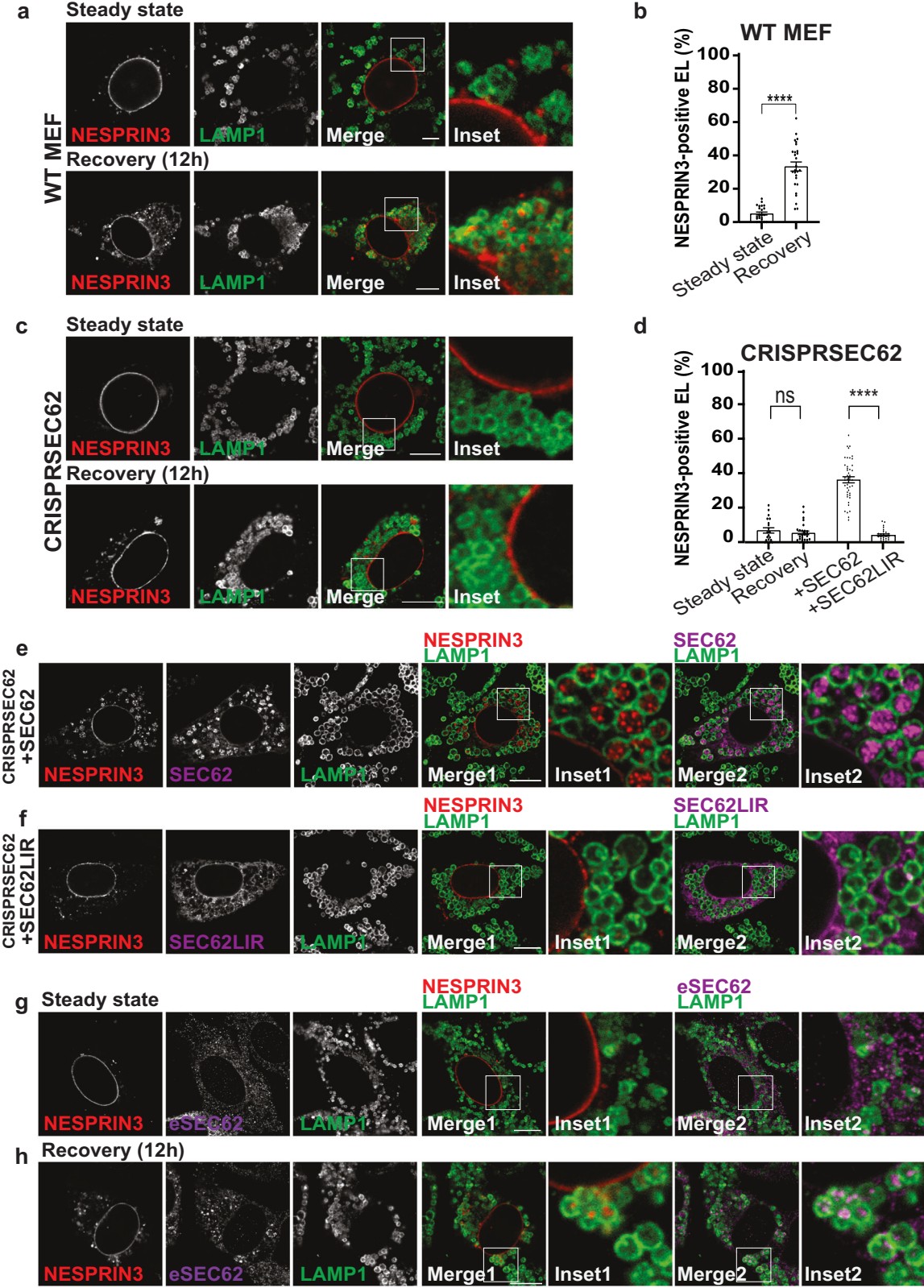

indirect immunofluorescence led to postulate a distribution of TMX4 in the ER and in the INM[68]. Our analyses by immunoelectron microscopy confirm the localization of endogenous TMX4 in the ER (arrowheads in Fig. 7g) but reveal that TMX4 distributes in the ONM (arrows in Fig. 7g). As a control, TEX264, a protein involved in ER-phagy[69,70] and in DNA repair[71,72] distributes in the ER, in the INM and in the ONM (Fig. 7h). Finally, TMX5 is an unconventional member of the

PDI superfamily, which has a cysteine-arginine-phenylalanine-serine (CRFS) catalytic site and is likely to engage clients in long-living disulfide-bonded complexes[64].

To assess the client specificity of TMX3, TMX4 and TMX5 in cellula, we generated mutant forms of the enzymes, where the last cysteine residue of the TMX's CXXC catalytic sites has been mutated to alanine. This mutation stabilizes the mixed disulfide that oxidoreductases

**Fig. 5 | SEC62 is the autophagy receptor involved in the lysosomal clearance of ONM subdomains. a** CLSM analyses of HALO-NESPRIN3α-positive ONM sub-domains delivery within LAMP1-positive endolysosomes in MEF at steady state (upper panels) or in MEF recovering from ER stress, 12 h after interruption of the pharmacologic treatment with CPA and exposure to 50 nM BafA1 (lower panels). Scale bars: 10 μm. **b** Quantification of (**a**) by LysoQuant[52]. *n* = 23 and 27 cells for steady state and recovery, respectively. *N* = 3 independent experiments. Mean ± SEM; unpaired, two-tailed *t*-test, ****P < 0.0001. **c** Same as (**a**) in MEF lacking the autophagy receptor SEC62[26,28]. Also refer to Supplementary Fig. 3. **d** Quantification of (**c**) *n* = 20 and 27 cells for steady state and recovery, respectively. *N* = 3 independent experiments, mean ± SEM; unpaired, two-tailed *t*-test, ns. *P* = 0.4279 and of (**e**) and (**f**) *n* = 44 and 27 cells for SEC62 and SEC62LIR, respectively. mean ± SEM; unpaired, two-tailed *t*-test, ****P < 0.0001. **e** Same as (**c**) in CRISPRSEC62 MEF back-transfected with SEC62. **f** Same as (**c**) in CRISPRSEC62 MEF back-transfected with SEC62 with a mutation in the LIR domain preventing LC3 association. **g** Same as (**a**) in MEF at steady state, to monitor the co-localization of HALO-NESPRIN3α and endogenous SEC62. **h** Same as (**g**) in MEF recovering from ER stress. Scale bars: 10 μm.

establish with client proteins. Consequently, client proteins remain disulfide-bonded to the oxidoreductase and can be identified upon co-immunoprecipitation and mass spectrometry analyses[63,73,74]. Briefly, V5-tagged versions of TMX3 and its trapping mutant TMX3$_{C56A}$, TMX4 and its trapping mutant TMX4$_{C67A}$ and of TMX5, a natural trapping mutant, were individually expressed in HEK293 cells. After immunoisolation from cell lysates with anti-V5 antibodies, the immunocomplexes containing TMX3-V5 (Fig. 8a, lanes 2 and 8), TMX3$_{C56A}$-V5 (lanes 3 and 9), TMX4-V5 (lanes 4 and 10), TMX4$_{C67A}$-V5 (lanes 5 and 11) or TMX5-V5 (lanes 6 and 12) were separated in non-reducing/reducing SDS-polyacrylamide gel, which was subsequently silver stained (Fig. 8a). TMX3$_{C56A}$, TMX4$_{C67A}$ and TMX5 are trapped in mixed disulfides with several cellular polypeptides (Fig. 8a, lanes 3, 5 and 6 respectively, red rectangles). The disulfide-bonded complexes are disassembled and disappear from this region of the gel when the immunocomplexes are run in the gel under reducing conditions (Fig. 8a, lanes 9, 11 and 12, yellow rectangles). As expected, the disulfide-bonded complexes are significantly less abundant in the lanes loaded with immunoisolates of cells expressing the active forms of TMX3 and TMX4, which rapidly release their clients (Fig. 8a, lanes 2 and 4, black rectangles).

The cellular proteins captured in disulfide-bonded complexes with TMX3$_{C56A}$, TMX4$_{C67A}$ and TMX5 were identified by mass spectrometry (Supplementary Table 1, Fig. 8b). Notably for the work presented in this submission, these analyses reveal endogenous NESPRIN proteins (i.e., NESPRIN2 (SYNE2) and NESPRIN1 (SYNE2)) as major TMX4 clients (Supplementary Table 1, Fig. 8b, red). NESPRIN proteins were not captured by the TMX1$_{C59A}$ trapping mutant that we investigated in a previous study[75], nor by the TMX3$_{C56A}$ trapping mutant (Supplementary Table 1, Fig. 8b), or by TMX5 (Supplementary Table 1, Fig. 8b) confirming the different client's subset for the four membrane-associated PDI family members.

Notably, despite the higher expression level in HEK293 cells of SUN1 (33.3 normalized transcripts per million (nTPM)) and SUN2 (54.7 nTPM) compared to NESPRIN1 (4 nTPM), NESPRIN2 (3.6 nTPM) and NESPRIN3 (1 nTPM)[76], SUN proteins were not trapped in mixed disulfides, showing that SUN proteins are not clients of the TMX4 reductase. All in all, TMX4 has reductase activity[67], distributes in the ONM (Fig. 7g) and endogenous NESPRIN proteins are TMX4 clients in living cells (Supplementary Table 1, Fig. 8b). Altogether, these results support a possible involvement of TMX4 in the modulation of the LINC complex via its reductase activity during ER stress.

### TMX4 forms mixed disulfides with NESPRIN3α

The resolution of an *inter*-molecular disulfide bond occurs upon nucleophilic attack of a reductase catalytic cysteine to a cysteine residue of one of the partners. Our data imply that TMX4 attacks (i.e., forms mixed disulfides with) NESPRIN proteins preferentially. To confirm that TMX4 preferentially engages NESPRIN proteins in mixed disulfides, HEK293 cells were mock-transfected (Fig. 9a, lanes 1, 5, 9), co-transfected with HALO-NESPRIN3α and GFP-SUN1 (Fig. 9a, lanes 2, 6, 10), with HALO-NESPRIN3α, GFP-SUN1 and TMX4$_{C67A}$-V5 (Fig. 9a, lanes 3, 7, 11), or with HALO-NESPRIN3α, GFP-SUN1 and TMX4-V5 (Fig. 9a, lanes 4, 8, 12). Cell lysates were separated in reducing SDS-polyacrylamide gels and processed for western blotting to verify the expression of the polypeptides. The PVDF membranes were probed with antibodies to the

V5 epitope (Fig. 9a, lanes 1–4) to confirm the expression of TMX4$_{C67A}$-V5 (TMX4*, lane 3) and TMX4-V5 (TMX4, lane 4). The ectopic expression of GFP-SUN1 was confirmed with antibodies to GFP (Fig. 9a, lanes 6–8). Likewise, the ectopic expression of HALO-NESPRIN3α was confirmed with antibodies to HALO (Fig. 9a, lanes 10–12).

The engagement of TMX4$_{C67A}$-V5 and TMX4-V5 in mixed disulfides was monitored by separation of the complexes immunoisolated from cell lysates with anti-V5 antibody in non-reducing gels and subsequent western blotting with anti-V5 antibody (Fig. 9b lanes 1–3). These analyses confirm the enrichment of mixed disulfides (MD) by the trapping mutant version of TMX4 (lanes 2 vs. 3, Fig. 9b). The V5-immunoreactive polypeptides shown with asterisks (Fig. 9b, lanes 2, 3) are also detected when the same PVDF membrane is probed with the anti-HALO antibody (Fig. 9b, circles in lanes 5, 6), revealing the TMX4 (lanes 2, 3) and the NESPRIN3α components (lanes 5, 6) of TMX4-V5-*SS*-HALO-NESPRIN3α mixed disulfides, respectively. In contrast, the MD immunoisolated from cell lysates with the anti-V5 antibody is not recognized by the anti-GFP antibody (Fig. 9b, lanes 7–9). These results confirm the data in cellula, which identify endogenous NESPRIN proteins, but not SUN proteins, as TMX4 clients (Fig. 8 and Supplementary Table 1).

TMX4-V5-*SS*-HALO-NESPRIN3α mixed disulfides are also immunoisolated from cell lysates with the anti-HALO antibody (Fig. 9c, lanes 10–15). In this case, the TMX4 component of the mixed disulfide is revealed by western blotting with the V5 antibody (polypeptide with asterisk in Fig. 9c, lanes 11, 12). The NESPRIN3α component of the mixed disulfides is revealed with the anti-HALO antibodies (Fig. 9c, lanes 14, 15, circles). In all cases, specificity is supported by the enhanced presence of disulfide-bonded complexes when these are trapped upon mutation of the TMX4 catalytic site (Fig. 9b, lanes 2 vs. 3 and 5 vs. 6, Fig. 9c, lanes 11 vs. 12 and 14 vs. 15).

To further support the preference of TMX4 to engage NESPRIN proteins, GFP-SUN1 (Fig. 9d, IP:GFP, lanes 1–3 and 7–9) or HALO-NESPRIN3α (Fig. 9d, IP:HALO, lanes 4–6 and 10–12) were immunoisolated from cell lysates with anti-GFP or anti-HALO antibodies, respectively. Western blotting with anti-V5 antibodies reveals the co-precipitation of GFP-SUN with a low amount of monomeric TMX4 (Fig. 9d, lanes 2, 3, arrows, and 8, 9) and the absence of GFP-SUN1-*SS*-HALO-NESPRIN3α mixed disulfides (Fig. 9d, lanes 2, 3). Instead, TMX4 abundantly co-precipitates with HALO-NESPRIN3α (Fig. 9d, lanes 5, 6 and 11, 12). Notably, most of the TMX4-associated immunoreactivity that co-precipitates with NESPRIN3α is in mixed disulfides (TMX4-V5-*SS*-HALO-NESPRIN3α), which run between the 130 and 175 kDa (MD, Fig. 9d, lane 5). MD disappears upon sample reduction to reveal the TMX4-V5 component upon WB with the anti-V5 antibody (Fig. 9d, lane 11). These results confirm the analyses in cellula, where endogenous NESPRIN proteins show up as clients of the TMX4 reductase (Fig. 8 and Supplementary Table 1)

### TMX4 is dispensable for SEC62-driven recov-ER-phagy, but intervenes in SEC62-regulated ONM-phagy

To explore the role of TMX4 in NE dynamics, its intracellular level was reduced upon RNA interference. The stress/recovery protocol shown in Figs. 3 and 5 was reproduced in MEF with normal levels of TMX4 upon transfection with a scrambled small interfering RNA

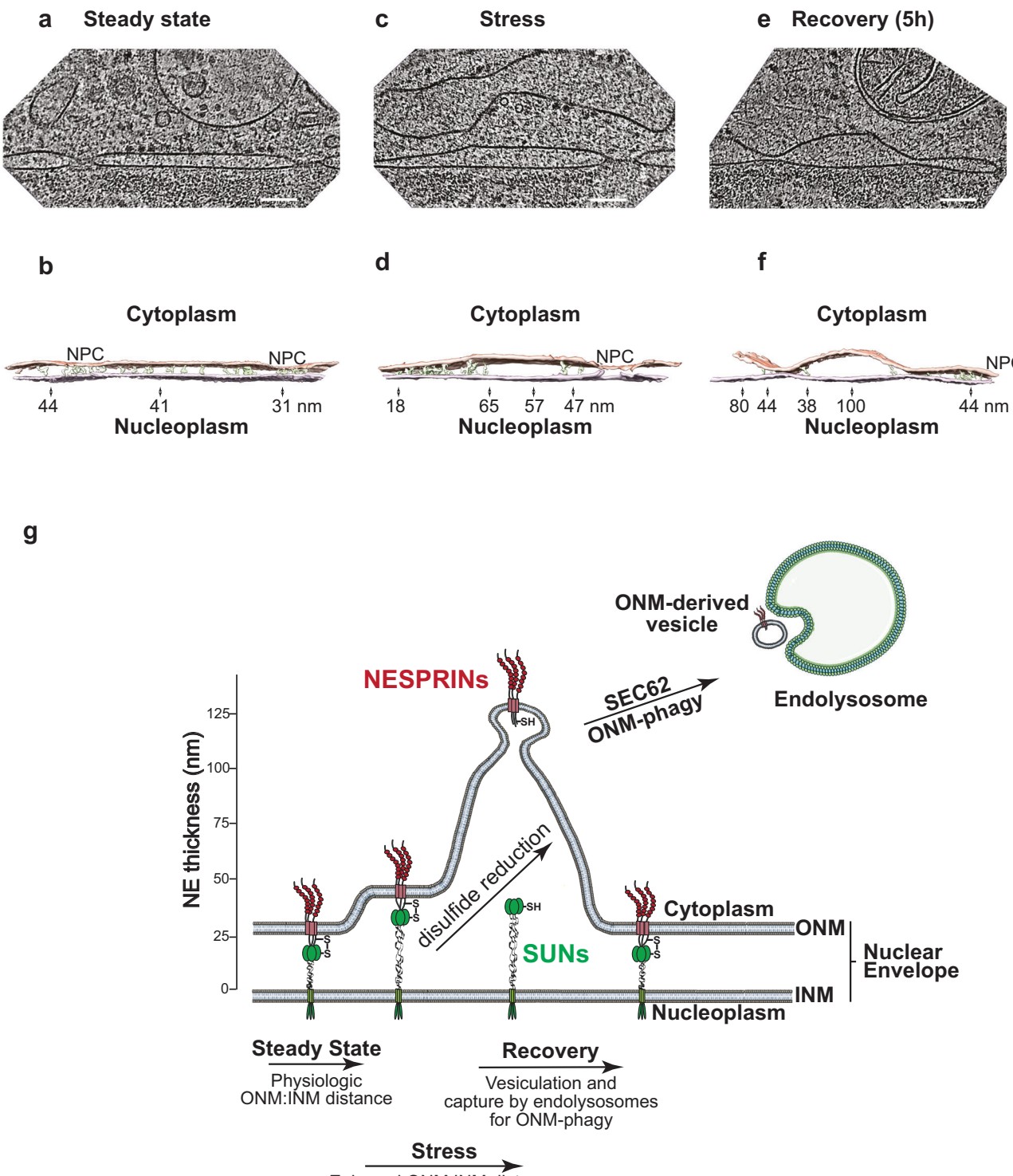

**Fig. 6 | Visualizing LINC complex filaments. a** Slice through cryo-electron tomogram of a NE in a FIB-milled MEF at Steady state. Scale bars represent 100 nm. **b** Isosurface representation of the corresponding segmented volume (ONM in salmon, INM in light purple). Filaments traced assisted by a density threshold mask on deconvoluted tomograms in Avizo in the perinuclear space are depicted in green (continuous from ONM to INM). Indicative distances between ONM and INM are given along the membrane. See tomogram 1, Supplementary Fig. 1a. **c** Same as (**a**) for cells under ER stress. **d** Same as (**b**). See tomogram 13, Supplementary Fig. 1b.

**e** Same as (**a**) for recovering cells. **f** Same as (**b**). See tomogram 30, Supplementary Fig. 1c. **g** Schematic representation of the NE before, during and after stress. α-helical coiled-coil perinuclear domains of SUN proteins can extend for a maximal length of 45–50 nm[21]. Enlargement of the PNS is allowed by the reduction of the disulfide that links SUN proteins in the INM with NESPRIN proteins in the ONM. (**g**) was partly drawn by using pictures from Servier Medical Art in Adobe Illustrator. Servier Medical Art by Servier is licensed under a Creative Commons Attribution 3.0 Unported License (https://creativecommons.org/licenses/by/3.0/).

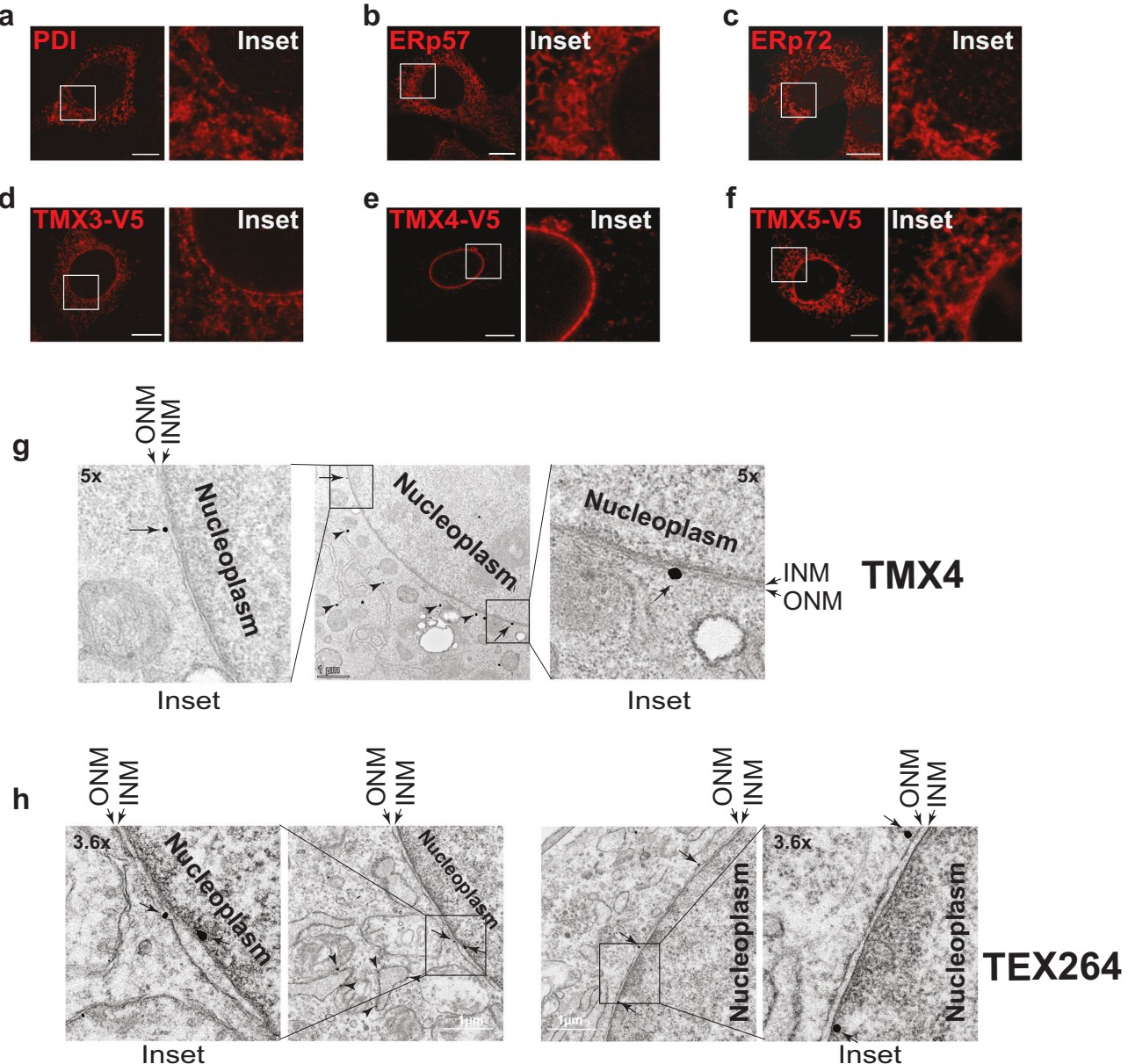

**Fig. 7 | Distribution of PDI family members in the ER and in the NE. a** CLSM analysis to monitor the subcellular distribution of endogenous PDI in MEF. Scale bars: 10 μm. **b** Same as (**a**) for ERp57. **c** Same as (**a**) for ERp72. **d** Same as (**a**) for ectopically expressed TMX3-V5. **e** Same as (**a**) for ectopically expressed TMX4-V5. **f** Same as (**a**) for ectopically expressed TMX5-V5. **g** RT-TEM micrograph showing immunogold labeling of endogenous TMX4 in the ER (arrowheads) and in the ONM of MEF (arrows). **h** RT-TEM micrograph showing immunogold labeling of endogenous TEX264 in the ER (arrowheads), in the ONM and INM of MEF (arrows). The CLSM experiments were performed once.

(Supplementary Fig. 5a, siSCR) and in MEF, where the level of TMX4 was reduced by 65% both at the mRNA (Supplementary Fig. 5a, siTMX4) and at the protein level (Supplementary Fig. 5b, siTMX4) upon transfection with a small interfering RNA targeting the TMX4 transcripts. In MEF transfected with siSCR, both SEC62 (Fig. 10a and Inset 1, Fig. 10c, siSCR) and HALO-NESPRIN3α (Fig. 10a and Inset 2, Fig. 10c, siSCR) are delivered within LAMP1-positive endolysosomes during recovery from ER stress. Silencing of TMX4 expression does not affect lysosomal delivery of SEC62 (Fig. 10b, and Inset 1, Fig. 10c, siSCR vs. siTMX4) but inhibits lysosomal delivery of HALO-NESPRIN3α (Fig. 10b, and Inset 2, Fig. 10c, siSCR vs. siTMX4). This observation hints at a mechanistic difference between lysosomal clearance of ER subdomains[26,28] and lysosomal clearance of ONM portions during recovery from ER stress. Both pathways involve the LC3 lipidation machinery but not autophagosome biogenesis (refs. 26,28 for ER subdomains, Fig. 3e–h for ONM portions) and rely on activation of the autophagy receptor SEC62 (ref. 26,28 for ER subdomains, Fig. 5c–e for ONM portions). However, only the lysosomal delivery of ONM, which requires the physical separation of the ONM from the INM, relies on TMX4 intervention (Fig. 10b, c). The failure to re-activate the ONM-phagy pathway by back-transfecting siRNA-resistant TMX4 in cells where the expression of the endogenous TMX4 is silenced led us to assess the consequences of TMX4 overexpression in cells recovering from acute CPA-induced stress. Strikingly, in wild-type MEF expressing recombinant TMX4, lysosomal delivery of SEC62 progresses normally (Fig. 10d and Inset 1), but lysosomal delivery of NESPRIN3α is substantially inhibited (Fig. 10d and Inset 2). Thus, silencing or overexpression of TMX4 does not affect SEC62-driven recov-ER-phagy. However, both silencing and overexpression of TMX4 substantially inhibit SEC62-driven ONM-phagy.

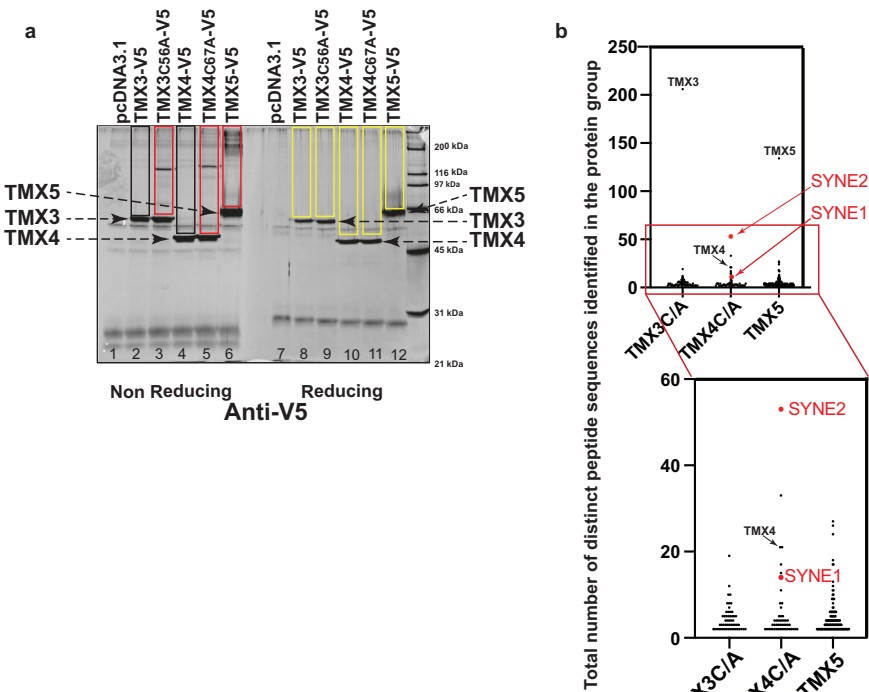

**Fig. 8 | Endogenous clients of TMX3, TMX4 and TMX5. a** Anti V5-antibody was used to immunoisolate from HEK293 cell lysates V5-tagged versions of TMX3 (lane 2), TMX3C56A (lane 3), TMX4 (lanes 4), TMX4C67A (lanes 5) and TMX5 (lanes 6). Part of the immunoisolates has been separate in non-reducing/reducing gel (lanes 1–6 and 7–12, respectively). The gel has been silver stained. Uncropped gel in Supplementary Fig. 6. The polypeptide bands in the black and red rectangles are disulfide-bonded complexes (mixed disulfides) associated with the respective TMX protein. The complexes disappear (yellow boxes), i.e., are disassembled, when the samples are run under reducing conditions (lanes 7–12). Part of the immunoisolates has been processed for mass spectrometry (see "Methods" section) to determine the composition of the mixed disulfides and identify the endogenous proteins trapped in mixed disulfides with the given TMX protein. **b** Graphical representation of Supplementary Table 1. Red dots show NESPRIN proteins.

## ER swelling is not transmitted to the NE in cells lacking TMX4

Next, we generated TMX4-KO MEF by CRISPR-Cas9 genome editing (Supplementary Fig. 5c). CRISPRTMX4 cells are unstable and must be examined at low passages, implying an important role of TMX4 in the maintenance of cultured cells' fitness and viability. Nevertheless, the NE of cells lacking TMX4 looks normal at steady state (Fig. 10e and insets, to be compared with Fig. 1b). Cells lacking TMX4 do respond to CPA-induced ER stress as shown by the upregulation of the conventional ER stress marker BiP/GRP78 at the transcript (Supplementary Fig. 5d) and at the protein level (Supplementary Fig. 5e). However, in contrast to wild type cells (Figs. 1, 4b–d, 6, Supplementary Fig. 4), the NE of CRISPRTMX4 cells does not significantly change morphology upon pharmacologic treatment with CPA (Fig. 10e vs. f). This confirms the role of the reductase TMX4 during the enlargement of PNS width resulting from perturbation of ER homeostasis.

## Discussion

Perturbation of ER homeostasis activates UPR that enhances protein and lipid synthesis to expand ER membranes and volume[24]. In this work, analyses of the NE ultrastructure by RT and cryo-EM reveal that the expansion of the mammalian ER lumen is transmitted to the contiguous PNS, whose width, at least in adherent cells and in cells characterized by increased mechanical tension[22,23] is maintained below the 50 nm[14,20,21] by LINC complexes formed by NESPRIN proteins in the ONM covalently linked, via *inter*molecular disulfide bonds, with SUN proteins in the INM. During pharmacologic perturbation of ER homeostasis, LINC complexes are disassembled, and the ONM forms bulges, which are not observed in cells lacking the ONM-membrane bound reductase TMX4. By trapping endogenous and ectopically expressed NESPRIN proteins

(but not SUN proteins) in mixed disulfides with TMX4, we show that the active site cysteine of TMX4 acts upon the NESPRIN cysteine residue involved in the NESPRIN-S-S-SUN intermolecular disulfide bond to disassemble the LINC complexes. For some ER-resident oxidoreductases, client specificity is well-established. For example, the BiP-associated P5 oxidoreductase assists the oxidative folding of clients of the ER chaperone BiP[74], the calnexin/calreticulin-associated ERp57 assists the folding of glycoproteins that engage the lectin chaperones calnexin and calreticulin[77,78]. In other cases, substrate-specificity may rely on the tissue-specific expression of the redox enzyme and its clients, as in the case of the testis-specific oxidoreductase PDILT[79,80]. For TMX4, the reason for the clear preference for NESPRIN proteins compared to the more abundant SUN proteins is unclear. The localization of the reductase in the ONM, which we established by immunogold EM, might play a role. Also unclear is how UPR induction may trigger the reductase activity of TMX4 required to disassemble LINC complexes. The recently reported modulation of TMX4 by Zdhhc6-driven palmitoylation offers an interesting possibility to activate TMX4 on demand[81]. It remains to be established if the TMX4-driven mechanisms that disassemble LINC complexes and expand the PNS upon perturbation of ER homeostasis also regulate nuclear envelope budding (NEB) events upon perturbation of protein folding in yeast[32] or the NPC-independent nucleoplasm to cytoplasm transport of macromolecules (e.g., ribonucleoprotein particles[36]), two cellular processes that require enlargement of the distance between the ONM and the INM. Future studies will also examine if the LINC complex disassembly machinery is hijacked by viruses such as Herpesviridae, whose capsid is produced in the nucleoplasm and must transit through the PNS for nuclear egress[34,35].

As previously reported for ER swelling[26–28,46–48], our results show that also the increase of the PNS width is reversible, and a few hours

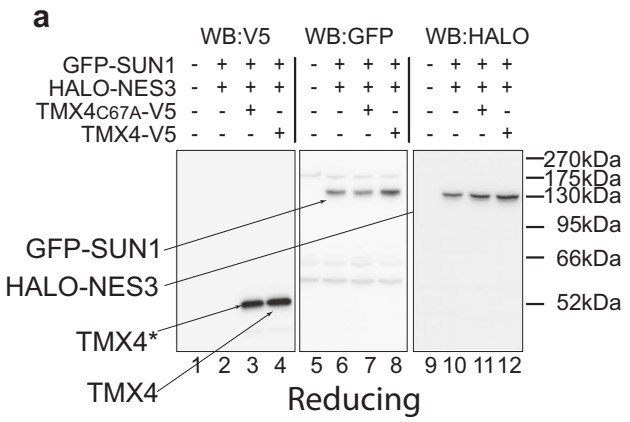

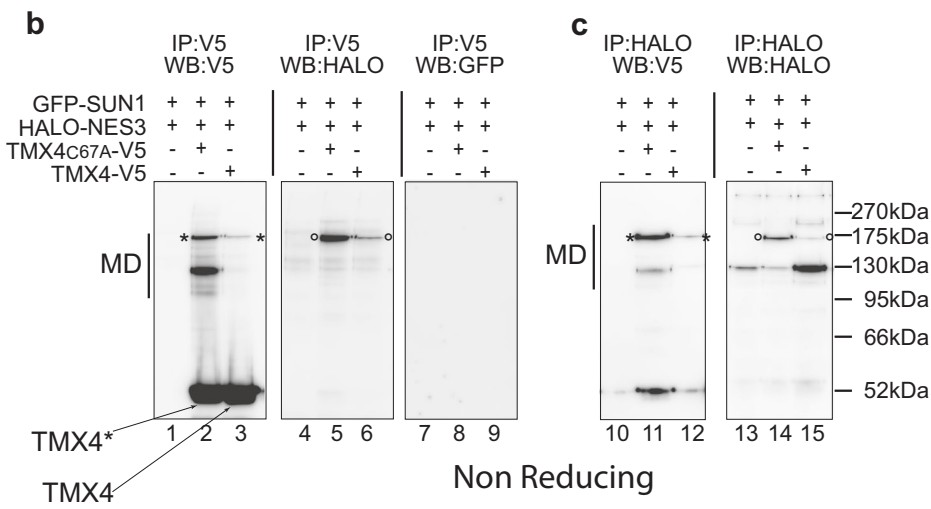

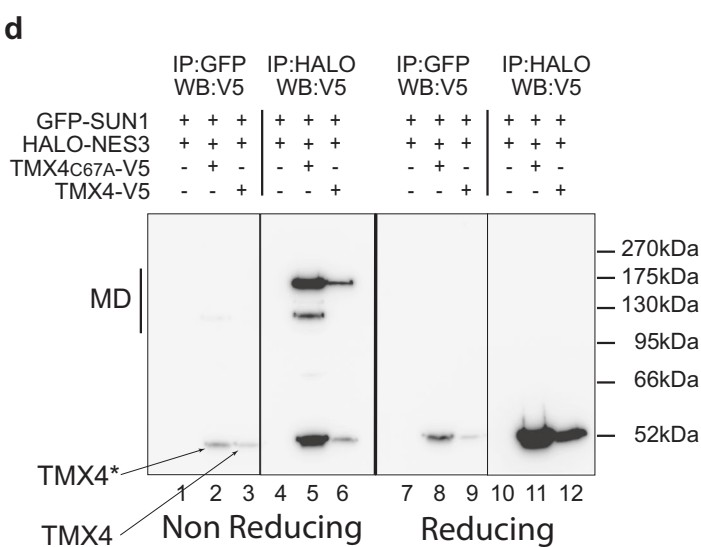

after interruption of the pharmacologic treatment, the physiologic ultrastructure of the NE is restored. This happens upon selective vesiculation of the ONM and direct capture of excess ONM by degradative endolysosomes via piecemeal *micro*-ONM-phagy. Like the recov-ER-phagy pathways that restore physiologic ER size and activity at the end of acute perturbations of ER homeostasis, what we here define as ONM-phagy involves the autophagy receptor SEC62 and the LC3 lipidation machinery. This is supported by genetic and morphometric analyses, which also show the dispensability of autophagosome biogenesis and intervention. The significant difference between ER and NE dynamics during perturbation of cellular homeostasis is that only the latter involves the reductase TMX4, which is activated during the pharmacologic treatment to transmit the stress-induced ER enlargement to the NE.

**Fig. 9 | In cellula assessment of TMX4 engagement in mixed disulfides with NESPRIN3α. a** Lysates of cells mock-transfected (lanes 1, 5, 9), expressing GFP-SUN1 and HALO-NESPRIN3α, TMX4C67A-V5 (lanes 2, 6, 10), GFP-SUN1 and HALO-NESPRIN3α (lanes 3, 7, 11), or TMX4-V5 GFP-SUN1 and HALO-NESPRIN3α (lanes 4, 8, 12) were separated in a reducing gel and transferred on PVDF membranes. Expression of the ectopic proteins was confirmed by WB with anti-V5 (lanes 1–4), anti-GFP (lanes 5–8), or anti-HALO antibodies (lanes 9–12). Uncropped blots in Supplementary Fig. 6. **b** The presence of TMX proteins (lanes 1–3), NESPRIN3α (lanes 4–6), or SUN1 (lanes 7–9) in mixed disulfides (MD) with TMX4C67A-V5 (lanes 2, 5, 8) or TMX4-V5 (lanes 3, 6, 9) immunoisolated from cell lysates with anti-V5

antibodies was monitored under non-reducing conditions by western blot with anti-V5, anti-HALO, or anti-GFP antibodies, respectively. **c.** The engagement of TMX proteins (lanes 10–12), or NESPRIN3α (lanes 13–15) in MD with HALO-NESPRIN3α immunoisolated from cell lysates with anti-HALO antibodies was monitored under non-reducing conditions by western blot with anti-V5 and anti-HALO antibodies, respectively. **d** The presence of TMX4 proteins in MD with GFP-SUN1 (lanes 1–3), or with HALO-NESPRIN3α (lanes 4–6) immunoisolated from cell lysates has been assessed by WB with anti-V5 antibodies under non-reducing conditions (or reducing conditions, lanes 7–12). WB is representative of at least three independent experiments.

## Methods

### Antibodies, expression plasmids
Commercial antibodies used in this study: Lamp1 (DSHB Hybridoma Product 1D4B deposited by J. T. August, Immunofluorescence (IF) 1:50), V5 (Western blot (WB) 1:5000 Thermo Fisher Scientific), GFP (WB 1:1000 IEM: 1:50 Abcam), TMX4 (WB 1:1000, IEM:1:20 Proteintech), TEX264 (IEM: 1:50 Novus), LC3B (IF: 1:50 Sigma), GAPDH (WB 1:30000 Millipore), KDEL (WB 1:1000 Stressgen), HaloTag (WB 1:1000 Promega), PDI (WB 1:100 Stressgen), Erp72 (WB 1:100 Stressgen). Alexa–conjugated secondary antibodies were purchased from Thermo Fisher Scientific (anti-rat 647, anti-rabbit 488), Invitrogen (anti-rabbit 568, anti-rabbit 405) and Jackson Immunoresearch (anti-mouse 488, anti-rabbit 657) (IF 1:300). The HRP-conjugated secondary antibodies were purchased from SouthernBiotech (anti-mouse, WB 1:20000) and Protein A HRP-conjugated were purchased from Invitrogen (WB 1:20000). The secondary Nanogold Fab anti-rabbit (H+L) and anti-mouse (H+L) antibodies were purchased from Nanoprobes (IEM: 1:100).

The HALO-Trap or GFP-Trap Agarose beads and V5-conjugated beads were purchased from ChromoTek and Sigma, respectively. The HaloTag ligands JF646 or tetram-ethylrhodamine (TMR) were purchased from Promega. Antibody against SEC62 (IF 1:100, IEM:1:50), CNX (WB 1:2000), FAM134B (WB 1:1000) and ERp57 (WB 1:100) were kind gifts from R. Zimmermann, A. Helenius, M. Miyazaki and T. Wileman, respectively. Plasmid encoding GFP-RAB7, GFP-SUN1, GFP-NESPRIN3α are kind gifts from T. Johansen, H. J. Worman and A. Sonnenberg, respectively. NESPRIN3α was subcloned in a pcDNA3.1 expression plasmid with the addition of an N-terminal HALO-Tag. The trapping mutants of TMX4 (TMX4$_{C67A}$-V5), TMX3 (TMX3$_{C53A}$-V5) and TMX5-V5 WT were synthesized by GenScript. The TMX3-V5 WT and TMX4-V5 WT were obtained from trapping mutants by site-directed mutagenesis.

### Cell culture, transient transfection, CRISPR/Cas9 genome editing, small interfering RNA silencing
MEF, HEK293 and NIH/3T3 cells were grown in DMEM with 10% fetal bovine serum (FBS) at 37 °C and 5% $CO_2$. Transient transfections were carried out using the JetPrime transfection reagent (PolyPlus) in accordance with the manufacturer's protocol. *Atg5* KO[56] and *Atg14* KO[82] MEF were kind gifts from N. Mizushima and T. Saitoh, respectively. CRISPRMOCK, CRISPRSEC62 and CRISPRFAM134B were generated by CRISPR/Cas9 genome editing in our lab as described in ref. 26. RNA interferences were performed in MEF plated at 50–60% confluence. Cells were transfected with scrambled small interfering RNA or small interfering RNA (siRNA) to silence TMX4 expression (5′-GCAUGGU GUUCUUACGUUUUtt-3′, 100 nM per dish, Silencer Select Pre-designed, Ambion). Cells were processed for immunofluorescence or for biochemical analyses after 48 h of transfection.

TMX4-KO MEF cells were generated as follows: the guideRNA-Cas9 plasmids, lentiCRISPRv2-puro system (Addgene52961) was obtained from Addgene (http://www.addgene.org). Cas9 target design tools were used to generate guide sequences (http://crispr.cos.uni-heidelberg.de/). All protocols and information can be found at the website https://www.addgene.org/crispr. The guide RNA target sequences were synthesized by Microsynth AG. Two annealed

oligonucleotides (5′-CACCGCTCGCAGCGGCAGCGGCCG-3′, 5′-AAACC GGCCGCTGCCGCTGCGAGC-3′ for murine TMX4 were inserted into the lentiCRISPRv2-puro vector using the *Bsm*BI restriction site. LentiV2- gRNA vector was transfected in MEF cells with JetPrime (Polyplus) according to the manufacturer's instructions. Cells were cultured in DMEM supplemented with 10% FBS. After 2 days of transfection, the medium was supplemented with 2 µg/ml puromycin (Invitrogen). Puromycin-resistant clones were picked after 10 days and gene KO was verified by WB.

### RNA extraction, polymerase chain reaction with reverse transcription (RT–PCR)
The extraction of RNA from MEF was performed with the GenElute Mammalian Total RNA Miniprep Kit (Sigma) according to the manufacturer's instructions. One microgram of RNA was used for cDNA synthesis with dNTPs (Kapa Biosystems), oligo(dT) and the Super-Script II reverse transcriptase (Thermo Fisher Scientific) according to the instructions of the manufacturer. For each qRT-PCR reaction, 10 µl of Power SYBR Green PCR Master Mix (Bimake), 0.4 µl of reference ROX dye and 7.6 µl of milliQ sterile water were added to 1 µl cDNA together with 1 µl of 10 µM forward and reverse primer mix (mTMX4 (I) fwd: (5′–3′): TTG AGT GGC CGC TTC TTT GT rev: (5′–3′): CCA GAC ATC GTT AGA GAG GCT; mTMX4 (II) fwd: (5′–3′): CAT CCT GCC AGC AGA CTG ATT rev: (5′–3′): GGC GGA ATA TCC CAT CTT TTG C), (mBiP fwd: (5′–3′): GAGTTCTTCAATGGCAAGGA rev: (5′–3′): CCAGTCAGATC AAATGTACCC) for the transcript of interest in 96-well reaction plate (MicroAmp Fast Optical 96-Well Reaction Plate with Barcode (0.1 ml), Applied Biosystems). The plate was vortexed and centrifuged. Samples were loaded as triplicates. Quantitative real-time PCR was performed using QuantStudio™ 3 Real-Time PCR System. The housekeeping gene actin was used as reference. Data were analyzed using the Quant-Studio™ Design & Analysis Software v1.5.5.

### Protocol to induce transient ER stress
To induce transient ER stress, cultured cells were exposed for 12 h to 10 µM CPA in DMEM, 10% FBS. For the recovery condition, CPA (Sigma) was washed out and incubation was prolonged in DMEM, 10% FBS up to 48 h. Cells were processed for biochemical or imaging analyses (as in refs. 26,28).

### Cell lysis
HEK293 cells plated on poly-L-lysine coated dishes were washed in cold phosphate-buffered saline (PBS) containing 20 mM N-ethylmaleimide (NEM) and were lysed in RIPA buffer (1% Triton X-100, 0.1% SDS, 0.5% sodium deoxycholate in HBS, pH 7.4, 20 mM NEM and protease inhibitors) and then collected by scraping, maintained for 20 min on ice and then post-nuclear supernatants (PNS) were collected by centrifugation at 10,000×*g* for 10 min.

### Proteins immunoprecipitation and Western blot
The nuclear lysates were incubated with HALO-Trap Agarose beads for 2 h at 4 °C to isolate HALO-NESPRIN3. After immunoprecipitation, samples were washed in 1 ml HBS 1x pH 7.4, 0.5% Triton two times. Beads were dried out and resuspended in sample buffer without DTT

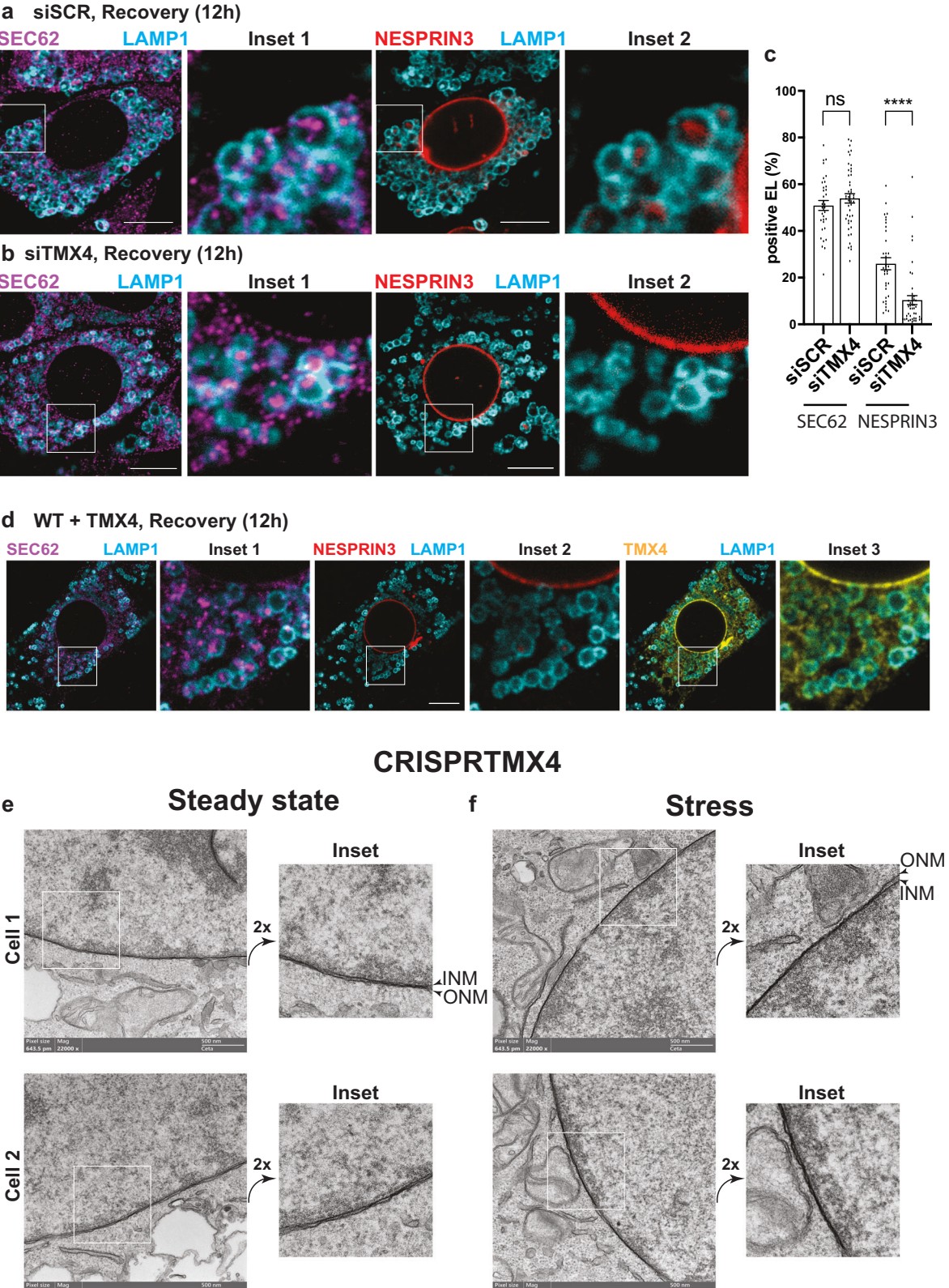

at 95 °C for 5 min. Then, half of the sample was reduced by the addition of 100 mM DTT. Samples were then boiled again for 5 min at 95 °C and loaded on 8% SDS-PAGE (polyacrylamide gel electrophoresis) for reducing and non-reducing protein separation. Proteins were transferred to polyvinylidene difluoride (PVDF) membranes with the Trans-Blot Turbo Transfer System (Bio-Rad). Membranes were blocked for 10 min with 8% (w/v) non-fat dry milk (Bio-Rad) in Tris-buffered saline (TBS)-Tween 1% and stained with primary antibodies (listed in "Methods") overnight, and for 45 min with HRP-conjugated secondary antibodies or Protein A HRP-conjugated. Membranes were developed using the Western bright ECL or Western bright QUANTUM (Witec) and signals were acquired with the FusionFX7 VILBER Witec using Fusion FX7 Edge software. Membrane stripping was performed using Re-Blot Plus Strong Solution (Millipore) following the manufacturer's

**Fig. 10 | Consequences of TMX4 silencing on NE dynamics. a** CLSM analyses of endogenous SEC62 (Inset1) and HALO-NESPRIN3α-positive ONM subdomains (Inset2) delivery within LAMP1-positive endolysosomes in MEF recovering from ER stress, 12 h after interruption of the pharmacologic treatment with CPA and exposed to 50 nM BafA1. Scale bars: 10 µm. **b** Same as (**a**), in cells where TMX4 expression has been silenced by RNA interference. Also, refer to Supplementary Fig. 5a, b. **c** Quantification of (**a**) and (**b**) in mock-treated cells and in cells with reduced expression of TMX4. $n = 34$ and 45 cells for mock-treated and TMX4 knockdown cells, respectively. $N = 3$ independent experiments, mean ± SEM; unpaired, two-tailed $t$-test, ns. $P = 0.3073$ for SEC62. $n = 34$ and 45 cells for mock-treated and with TMX4 knockdown cells, respectively. $N = 3$ independent experiments, mean ± SEM; unpaired, two-tailed $t$-test, ****$P < 0.0001$ for NESPRIN3α. **d** Same as (**a**) in wild-type MEF overexpressing TMX4. Representative of three independent experiments. Scale bars: 10 µm. **e** RT-TEM micrographs of NE of two different CRISPRTMX4 MEF at steady state. Scale bars: 500 nm. **f** Same as (**d**) 2 different CRISPRTMX4 MEF during pharmacologically-induced ER stress.

instructions before repetition of the protocol for the detection of other antigens.

### Silver staining
Polyacrylamide gels were fixed in a 40% EtOH, 10% Acetic Acid solution for 1–4 h at room temperature, rinsed two times for 20 min in 30% EtOH and then soaked for 20 min in $H_2O$. After 1 min in 0.02% $Na_2S_2O_3$ and three washes in $H_2O$, the gels were incubated for 20 min in 0.2% $AgNO_3$ and then rinsed three times in $H_2O$. Development was performed with a freshly prepared developing solution containing 3% $Na_2CO_3$ and 0.05% formaldehyde. Development was stopped by rinsing the gels in $H_2O$ and then in 5% Acetic Acid.

### Confocal laser scanning microscopy (CLSM)
MEF were seeded on alcian blue-treated glass coverslips and transiently transfected with JetPrime reagent according to the manufacturer's protocol. Cells were then treated with 10 µM CPA for 12 h and, on CPA removal, incubated for 12 h with 50 nM BafA1 (Calbiochem), or DMSO (Sigma) and with 100 nM TMR HaloTag ligand (Promega). Following two PBS washes, cells were fixed with 3.7% formaldehyde diluted in PBS for 20 min at RT. Cells were permeabilized with 0.05% saponin, 10% goat serum, 10 mM HEPES and 15 mM glycine (PS) for 15 min. Cells were incubated with the primary antibodies diluted 1:100 in PS for 120 min, washed two times in PS, and then incubated with Alexa Fluor-conjugated secondary antibodies diluted 1:300 in PS for 45 min. Cells were rinsed three times with PS and once with water. Afterward, cells were mounted with Vectashield (Vector Laboratories) with 4′,6-diamidino-2-phenylindole (DAPI).

Leica TCS SP5 or Leica Stellaris SP8 microscope with a Leica HCX PL APO lambda blue 63.0 × 1.40 OIL UV objective was used to acquire confocal images. The acquisition software Leica LAS X was used.

The quantifications of HALO-NESPRIN3α, GFP-SUN1, SUN2, SEC62, SEC62-GFP or SEC62LIR-GFP positive lysosomes per cell were executed with LysoQuant, an unbiased and automated deep learning tool for fluorescent image quantification, which is freely available (https://www.irb.usi.ch/lysoquant/)[52]. Image processing was also performed with Fiji/ImageJ and Adobe Photoshop.

### Live imaging
NIH3T3 cells were seeded on glass bottom Mattek 35 mm dishes (MatTek #1.5 coverslips) and transiently transfected with JetPrime reagent according to the manufacturer's protocol for GFP-RAB7 and HALO-NESPRIN3. Cells were treated with 10 µM CPA for 12 h and, on CPA removal, incubated for 5 h with 100 nM BafA1, and with 100 nM Janelia Flour 646 HaloTag ligand (Promega).

Movies were recorded with a Leica TCS SP5 confocal microscope with a Leica HCX PL APO lambda blue 63.0 × 1.40 OIL UV objective was used to acquire confocal images at 37 °C, 5% $CO_2$. Excitation was performed with 488 nm and 633 nm laser lines and fluorescence light was collected in ranges 493–625 nm and 637–791 nm respectively with pinhole 1 AU. The pictures were taken every 1 s with a pixel size of 63 nm and line average 3. Acquisitions were then cleaned with a median filter and movie processing was performed using Fiji/ImageJ.

### Mass spectrometry
HEK293 cells were transiently mock-transfected (pcDNA3.1) or transfected with TMX3-V5, TMX3$_{C53A}$-V5, TMX4-V5, TMX4$_{C64A}$-V5 and TMX5-V5. Sixteen h after transient transfection, cells were lysed in RIPA buffer and the PNS (post-nuclear supernatant) was collected. The PNS was double immunoprecipitated using anti-V5 conjugated beads. After washing, the beads used for immunoprecipitation were extracted by boiling for 5 min in 60 µl of sample buffer. The supernatant was split into two aliquots, one of which was reduced by adding DTT to a final concentration of 10 mM and incubating for 5 min at RT. Reduced and nonreduced samples were separated in a gel that was silver stained (Fig. 8a) or migrated on separate 10% polyacrylamide gels for a total distance of 4.0 cm. After Coomassie-blue staining, gel lanes loaded with the trapping variants of the three TMX proteins were excised in 4 bands from the loading point down to the 50 kDa marker and digested in-gel as described[83,84]. Extracted tryptic peptides were dried and resuspended in 0.05% trifluoroacetic acid, 2% (v/v) acetonitrile.

### LC-MS/MS and data analysis
Data-dependent LC-MS/MS analysis of extracted peptide mixtures after digestion with trypsin was carried out on a Q-Exactive Plus mass spectrometer (Thermo Fisher Scientific) interfaced to a nanocapillary HPLC (Dionex RSLC 3000). Peptides were separated on a custom-packed C18 reversed-phase column (75 µm ID × 45 cm, 1.8 µm particles, Reprosil Pur, Dr. Maisch), using a gradient from 4 to 76% acetonitrile in 0.1% formic acid (total time: 65 min). Full MS survey scans were performed at 70,000 resolution. In data-dependent acquisition controlled by Xcalibur software (Tune 2.9, Thermo Fisher Scientific), the 10 most intense multiply charged precursor ions detected in the full MS survey scan were selected for higher energy collision-induced dissociation (HCD, normalized collision energy NCE = 27%) and analysis in the orbitrap at 17′500 resolution. The window for precursor isolation was of 1.5 m/z units around the precursor and selected fragments were excluded for 60 s from further analysis. Collections of peptide tandem mass spectra were searched using Mascot 2.6.2 (Matrix Science, London, UK) against the January 2019 release of the SWISSPROT database limited to human taxonomy (20′762 sequences), together with a custom-built database containing common contaminants. Carbamidomethyl (Cys) was defined as fixed modification while oxidation (Met) and N-terminal protein acetylation were specified as variable modifications. The software Scaffold (version 4.9.0, Proteome Software Inc.) was used to validate MS/MS-based peptide (minimum 90% probability[85]) and protein (min 95% probability[86]) identifications, perform dataset alignment as well as parsimony analysis to discriminate homologous hits. Identifications were filtered for a maximum FDR (false discovery rate) of 1% against a decoy database. Analysis was performed once and then confirmed by in vitro analysis shown in Fig. 9.

### Statistical analyses and reproducibility
The legend of the figures indicates the number ($n$) of cells analyzed and the number $N$ of independent experiments (biological replicates). Statistical analyses were performed using GraphPad Prism 9 software. Unpaired, two-tailed $t$-test were used to assess statistical significance. $P$-values are given in the figure legends; ns. $P > 0.05$; *$P < 0.05$; **$P < 0.01$; ***$P < 0.001$; ****$P < 0.0001$.

Experiments for RT-TEM (Figs. 1, 2, 4 and 7) were repeated at least two times, up to five times, by four different operators. Experiment for 24 h recovery was also performed with consistent results (not shown in the paper). For Fig. 10e and f results of one experiment were analyzed in RT-TEM. Data were confirmed by RNA interference (Fig. 10a and b). Additional, independent experiments were analyzed by CET.

## Cryo-electron tomography (CET): Grid preparation
MEF were seeded on R2/2 holey carbon on gold grids (Quantifoil or Protochips) coated with fibronectin in a glass bottom dish (Mattek or Ibidi) and incubated for 12 h. For the stress condition, cells were then incubated for 12 h with 10 μM CPA in DMEM+10% FBS. For the recovery condition, CPA was washed out and cells were incubated in DMEM +10% FBS for another 5 h. Grids were mounted to a manual plunger, blotted from the back for ~10 s and plunged into liquid ethane.

## CET: lamellae preparation
Lamellae were prepared using an Aquilos FIB-SEM system (Thermo Fisher Scientific). Grids were sputtered with an initial platinum coat (10 s) followed by a 10 s gas injection system (GIS) to add an extra protective layer of organometallic platinum. Samples were tilted to an angle of 15° to 22° and 11-μm-wide lamellae were prepared. The milling process was performed with an ion beam of 30 kV energy in three steps: (1) 500 pA, gap 3 μm with expansion joints, (2) 300 pA, gap 1 μm, (3) 100 pA, gap 500 nm. Lamellae were finally polished at 30–50 pA with a gap of 200 nm.

## CET: data acquisition
A total of 66 tilt series were acquired on a Talos Arctica (Thermo Fisher Scientific) operated at an acceleration voltage of 200 kV and equipped with a K2 summit direct electron detector and 20 eV slit energy filter (Gatan). Images were recorded in movies of 5–8 frames at a target defocus of 4–6 μm and an object pixel size of 2.17 Å. Tilt series were acquired in SerialEM[87] using a grouped dose-symmetric tilt scheme[88] covering a range of ±60° with a pre-tilt of ±10° and an angular increment of 3°. The cumulative dose of a series did not exceed 80 e-/Å². On lower quality lamellae from CPA recovery condition, lower magnification tilt series were also acquired at an object pixel size of 4.47 Å, for membrane segmentation and illustration purposes (not used for distance measurements nor filament segmentation).

## CET: tomogram reconstruction
Movie files of individual projection images were motion-corrected with MotionCor2[89] and combined into stacks of tilt series using a Matlab script. The combined stacks were dose corrected in Matlab[90] and aligned using patch tracking in IMOD[91]. Full tomograms were reconstructed by weighted back projection at a pixel size of 13.06 Å. Ice thickness was determined manually and was found to be <200 nm for all tomograms.

## CET: ONM to INM distance measurements
A density threshold mask was applied in Avizo software (Thermo Fisher Scientific) to highlight only stronger densities. ONM and INM were then traced for each tomogram with the brush tool. The exported labels were submitted to surface morphometrics[92] to create a triangular mesh and measure the intra-membrane distance along the surfaces. Results were plotted in Python using matplotlib.

## CET: filament segmentation
For each dataset, the highest quality tomogram, corresponding to the thinner lamellae, was used for the analysis of the filaments in the perinuclear region. Deconvolution was applied using tom_deconv in Matlab (https://github.com/dtegunov/tom_deconv) and the resulting tomograms were imported into Avizo software (Thermo Fisher

Scientific). A density threshold mask was applied to highlight only stronger densities to guide subsequent segmentation steps. Assisted by the density threshold mask, the continuous pieces of density between ONM and INM visible in the perinuclear space in each z slice were traced with the brush tool. The resulting labels were exported for visualization in UCSF Chimera[93].

## Room temperature-transmission electron microscopy (RT-TEM)
MEFs were seeded on alcian blue-coated glass coverslips and fixed in double-strength fixative into the media (4% PFA EM grade, 5% GA in Na-cacodylate buffer 0.1 M, pH 7.4) for 20 min at RT. After removing the mixture, cells were incubated with single-strength fixative (2% PFA, 2.5% GA in Na-cacodylate buffer 0.1 M, pH 7.4) for 3 h at RT. After several washes in cacodylate buffer, cells were post-fixed in 1% osmium tetroxide (OsO₄), 1.5% potassium ferricyanide (K₄[Fe(CN)₆]) in 0.1 M Na-cacodylate buffer for 1 h on ice, washed with distilled water (dH₂O) and enbloc stained with 0.5% uranyl acetate in dH₂O overnight at 4 °C in the dark. Samples were rinsed in dH₂O, dehydrated with increasing concentrations of ethanol, embedded in Epon resin and cured in an oven at 60 °C for 48 h. Ultrathin sections (70–90 nm) were collected using an ultramicrotome (UC7, Leica microsystem, Vienna, Austria), stained with uranyl acetate and Sato's lead solutions and observed in a Transmission Electron Microscope Talos L120C (FEI, Thermo Fisher Scientific) operating at 120 kV. Images were acquired with a Ceta CCD camera (FEI, Thermo Fisher Scientific). Image analysis was performed using Microscopy Image Browser[45].

## RT-TEM: immunogold electron microscopy
Cells were fixed in 4% PFA EM grade and 0.2 M HEPES buffer for 1 h at RT or in Periodate-lysine-paraformaldehyde (PLP) for 2 h at RT. After three washes in PBS, cells were incubated 10 min with 50 mM glycine and blocked 1 h in blocking buffer (0.2% bovine serum albumin, 5% goat serum, 50 mM NH₄Cl, 0.1% saponin, 20 mM PO₄ buffer, 150 mM NaCl). Staining with primary antibodies and nanogold-labeled secondary antibodies (Nanoprobes) were performed in a blocking buffer at RT. Cells were fixed for 30 min in 1% GA and nanogold was enlarged with gold enhancement solution (Nanoprobes) according to the manufacturer's instructions. Cells were post-fixed with OsO₄ and processed as described for conventional EM. Images were acquired with Talos L120C TEM (FEI, Thermo Fisher Scientific) operating at 120 kV. Images were acquired with a Ceta CCD camera (FEI, Thermo Fisher Scientific) using Velox 3.6.0 (FEI, Thermo Fisher Scientific).

## RT-electron tomography (RT-ET)
For electron tomography, 130–150 nm thick sections were collected on formvar-coated copper slot grids and gold fiducials (10 nm) were applied on both surfaces of the grids. The samples were imaged in a 120 kV Talos L120C TEM (FEI, Thermo Fisher Scientific). Single or dual tilted image series (+60/−60) were acquired using Tomography 4.0 software (FEI, Thermo Fisher Scientific). Tilted series alignment and tomography reconstruction were done with the IMOD software package[94]. Segmentation and 3D visualization were done with Microscopy Image Browser[45] or IMOD software packages.

## RT-TEM: measurement of INM-ONM distance
ONM and INM were manually segmented in Microscopy Image Browser (MIB)[45]. Using the cell wall thickness plugin of MIB, the distance between the INM and the ONM was measured. Results were plotted in GraphaPad PRISM 9.

## Reporting summary
Further information on research design is available in the Nature Portfolio Reporting Summary linked to this article.

## Data availability

The source data are provided with this paper as a Source Data file. The mass spectrometry proteomics data have been deposited to the ProteomeXchange Consortium via the PRIDE partner repository with the dataset identifier PXD041156 and https://doi.org/10.6019/PXD041156. Source data are provided with this paper.

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

## Acknowledgements

We thank the members of Molinari's laboratory for the discussions and critical reading of the manuscript. We also thank Euro-BioImaging (www.eurobioimaging.eu) for providing access to imaging technologies and services via the Italian Node (ALEMBIC, Milan-Italy), Mihajlo Vanevic, Benjamin A. Barad, Miguel R. Leung and Gonzalo Obal for help with data processing and Andreas F. Sonnen for initial help with FIB milling. We are grateful to Stuart C. Howes and Menno Bergmeijer for cryo-EM support as well as to Mariska Gröllers-Mulderij for cell culture support. M.M. is supported by the ALPHA-1 Foundation Research Grant (ID: 681136), the Foundation for Research on Neurodegenerative Diseases, the Swiss National Science Foundation (SNF, 310030_184827/2), the Eurostar (E! 113321–CHAPERONE), Innosuisse (35449.1 IP-LS), and the Comel and Gelu Foundations. The work was also supported by the European Research Council under the European Union's Horizon 2020 Program (ERC Consolidator Grant Agreement 724425-BENDER) and the Nederlandse Organisatie voor Wetenschappelijke Onderzoek (Vici 724.016.001 to F.F. and Veni 212.152 to J.F.).

## Author contributions

M.K.K. and J.F. performed the experiments, and the biochemical, imaging, transcriptional and quantitative analyses; C.G. and T.S. prepared DNA constructs and performed qPCR and biochemical analyses; D.M. assisted in CSLM analyses and set up and adapted the LysoQuant program. A.R. performed the analyses with the RT-TEM. J.F. and F.F. performed the FIB-CET analyses. M.M. designed the study, analyzed data, and wrote the manuscript. All the authors discussed the results and the manuscript.

## Competing interests

The authors declare no competing interests.
