## [Peer Review File · Nature Communications]

TMX4-driven LINC complex disassembly and asymmetric autophagy of the nuclear envelope upon acute ER stressREVIEWER COMMENTS

Reviewer #1 (Remarks to the Author):

Kucinska et al report several exciting data. There is a lot of potential here and if done well, these experiments would be appropriate for a high impact manuscript to be read by a general audience. However, as presented here, this work is far from ready and does not reach its potential. The short format does not represent the data well. The paper is very poorly written. It is inaccessible to the non-specialist reader. There is little to no rationale for the experiments. A proper introduction and discussion would greatly improve this manuscript and make it more impactful. In addition, the figures are poorly done. They are too small and too contrasted. The live fluorescence images appear overly adjusted to remove background. The EM data are also too small. More 3-D moves of the tomograms would better represent the data. I don't see legends for the movies. Movies corresponding to Figure 3 would be helpful, as these data are central to the model and conclusions. This paper has potential, and I wish I could see more of the data.

1. In Fig 2C, SUN1 is looked at. SUN2 would be the more canonical SUN. SUN2 needs to be examined.
2. The tomograms in 1D and SF1A are impressive, and I'm happy the n is large. However, I feel like there is a missed quantification opportunity. The data in SF1 should be averaged somehow. Also, the way the blue boxes are drawn is confusing. The model at the top of SF1 is a model and should not be presented as fact. Remove the cartoon from Sup Fig 1.
3. The data in Fig 1 are key to the paper. However, NESPRIN3 is overexpressed from a CMV promoter in pCDNA3.1. It is highly likely that the vast overexpression of NESPRIN3-HALO is not interacting with a SUN protein. If there is more NESPRIN3 than SUN, than it could more easily be pulled away from the ONM.
4. Related to the above point, look at endogenous NESPRIN3 and 1 and/or 2 also. Seeing multiple endogenous ONM components moving toward the lysosomes would support your model more strongly.
5. FRAP experiments could help determine the dynamics of NESPRIN3. This could be confirmed by mutating the Cys residues, knocking down SUN, and knocking down TMX4. Overexpressed NESPRIN3 is likely to be more dynamic.
6. Which isoform of NESPRIN3 is examined? The isoform that interact with plectin or not? This could complicate analyses.
7. In the models here, TMX4 is predicted to be specific to the ONM. But in Cheng et al 2019 (ref 34), digitonin experiments suggest TMX4 is enriched at the INM. If it's at the INM, how could it reach the SUN-KASH disulfide bond? Immuno-EM like those done for Sec62, should be done for TMX4 before you can claim its enriched at the ONM. If it were at the ONM, how would it get there?
8. In Fig 4D, a siRNA resistant construct should be used to rescue the TMX4 phenotype.

9. Intermediate disulfide bonds are seen between TMX4 and NESPRINS by multiple assays. But no TMX4-SUN disulfide bonds were looked for in the gels in 4B, and they were not seen in the mass spec. Why are there no SUN-TMX4 intermediates? Also, why wasn't NESPRIN3 pulled down in the mass spec? Can you do the opposite, IP with NESPRIN3 and see if you specifically pull down TMX4?

10. Many references are wrong. For example, references 3-5 are cited throughout for LINC complexes regulating the width of the NE. These references only hypothesize about the width of the NE. Crisp et al 2006 and Cain et al 2014 are the only two places I know of that attempt to answer this question, and neither does so satisfactorily. The idea that LINC complexes regulate set and maintain NE width is still a model. References 6 and 7 have nothing to do with NESPRIN or LINC. Ref 21 is a review, instead it should cite a paper showing what the two sentences are claiming about NESPRIN3 and SUN1. I suspect there are many more examples.

11. Antibody verification is required for commercial antibodies against sSe62 and NESPRIN3. Or we need to be pointed to where the antibodies used have been verified to be specific.

12. The sentence on line 238 is overblown. This is already an active field!

Reviewer #2 (Remarks to the Author):

In this manuscript, the authors analyze the morphology of the nuclear envelope when mammalian cells are treated with the ER stress-inducing compound CPA. They discovered that ER stress causes the swelling of perinuclear space and during recovery, portions of ONM undergo autophagic removal in a manner depending on the recov-ER-phagy receptor SEC62. Interestingly, the authors find that the transmembrane reductase TMX4 specifically forms mixed disulfide bonds with NESPRIN proteins, which are the ONM components of LINC complexes. Knocking down TMX4 reduces the autophagic delivery of NESPRIN3 to endolysosomes. This work identifies a new ER-stress induced cellular process and unveils a new mechanism regulating LINC complexes.

Major points:

I think the identification of TMX4 as a reductase disassembling LINC complexes is the most exciting finding of this manuscript. Unfortunately, the loss-of-function phenotype generated using siRNA is rather weak and unconvincing, presumably because the knockdown is incomplete. Is it possible for the authors to generate TMX4 knockout cell lines? In addition, it will be interesting to analyze whether the loss of TMX4 prevents the increase of the width of the perinuclear space during ER stress.

Minor points

(1) In the manuscript, the authors used both the term "periplasmic space" (e.g. in the abstract) and the term "perinuclear space" (e.g. in Figure 3B legend). I think "perinuclear space" is a better term and does not cause confusion with bacterial periplasmic space. The authors may want to consider using "perinuclear space" throughout the manuscript.

(2) In the abstract, "Here we report that expansion of the mammalian ER upon homeostatic perturbations is transmitted to the NE, where the ONM forms large bulges." I think "ONM forms large bulges" is a potentially misleading way to describe the morphological changes. It seems to me that the increase of the width of the perinuclear space is a more prominent phenomenon than the formation of ONM bulges. The authors may want to consider changing "the ONM forms large bulges" to "the width of the perinuclear space increases".

(3) In Fig. 3G, why are mixed disulfide bonds only drawn between the reductase and NESPRIN proteins but not between the reductase and SUN proteins?

(4) Page 2, line 49. "Analyses of 17 tomograms reveals that at steady state". "reveals" should be "reveal".

Reviewer #3 (Remarks to the Author):

In the manuscript by Kucińska, Fedry et al., the authors describe an ER-phagy pathway specific for the clearance of portions of the outer nuclear membrane. They ultrastructurally describe changes in ER morphology in different cell culture systems upon inhibiting the sarco/ER calcium pump. By various EM methods including FIB/SEM cryo-ET, they show that after treatment and during recovery the distance between the inner and outer nuclear membrane increases. At these sites Nesprin3 dissociates from the LINC complex and is found in LAMP1-positive endolysosomes as shown by fluorescence microscopy experiments. They link this degradation to the selective autophagy receptor Sec62, since deletion of this protein inhibits Nesprin3 targeting to endolysosomes. Finally, their experiments suggest that the dissociation of Nesprin3 from the LINK complex is mediated by the oxidoreductase TMX4. Knockdown of TMX4 indeed blocks Nesprin3 accumulation in endolysosomes but does not affect Sec62 accumulation.

The identification of a ERphagy pathway specific for the degradation of the outer nuclear membrane is an exciting finding and the manuscript is well written. However, additional experiments are needed to strengthen the conclusion that a Sec62 dependent macroautophagy pathway is responsible for outer nuclear membrane degradation.

Major points:

1.) In order to establish Sec62 as the receptor for this selective autophagy pathway the authors should replace Sec62 by the LIR mutant to separate the translocon function of Sec62 from its involvement in autophagy.

2.) It is necessary to further strengthen the point of a novel macroautophagy pathway by characterizing it in more detail. Does this pathway lead to Sec62 dependent Nesprin3 degradation? Given that Atg8ylation is a sensor for membrane stress and not necessarily involved in macroautophagy the authors should test other autophagy mutants which are not involved in Atg8 lipidation. Do autophagy components such as LC3 colocalize with Nesprin3 at the nuclear envelope during vesicle formation?

3.) From many of the cryo-ET tomograms it is not clear that the vesicles described to pinch off the ONM are next to autophagic structures or endolysosomes as shown by fluorescence microscopy. Only Figure 2M shows this. Are these described vesicles really the target of autophagy, since potentially other pathways such as COPI/II could act on the nuclear envelope (PMID: 8548805). For this the authors should perform CLEM experiments using Nesprin3 to guide tomogram acquisition.

4.) CLEM experiments showing Nesprin3 positive vesicles within autophagosomes would be another strong indication of the existence of an ONM specific autophagy pathway.

Additional points:

4.) The images in Figure 3A do not allow to see the densities the authors have segmented in Figure 3B. Movies of the tomograms should be provided. In addition, it is unclear how many tomograms were analyzed? It would be good to do the segmentation (if possible) in an automated fashion as for example described in this publication (PMID: 31907446).

5.) One would expect that the Nesprin3 signal always colocalize with the Sec62 signal inside of endolysosomes. Is this the case? Would be nice to show a merged image between the Nesprin and Sec62 signal in Figure 4C.

6.) For Mov 1: It would be nice to have the movie played without the segmentation first to access the raw data.

7.) Figure S3A is not very clear. Vesicles in the inlay are only to imagine. The authors should enhance the brightness/contrast of this image and mark the Nesprin3 positive vesicles.

8.) LINE 113 – 116: “In yeast, selective autophagy of nuclear components (i.e., nucleus-derived double membrane vesicles, NE domains containing specific nucleoporins, nuclear pore complexes) has mainly been studied as a catabolic process induced upon nutrient deprivation and it involves the autophagy receptor Atg39.” In the case of NPCs there are additional receptors suggested in the literature (PMID: 32029894; PMID: 32453403)

Point-by-point response to the reviewers' comments, reproduced verbatim

Reviewer #1 (Remarks to the Author):

Kucinska et al report several exciting data. There is a lot of potential here and if done well, these experiments would be appropriate for a high impact manuscript to be read by a general audience. However, as presented here, this work is far from ready and does not reach its potential. The short format does not represent the data well. The paper is very poorly written. It is inaccessible to the non-specialist reader. There is little to no rationale for the experiments. A proper introduction and discussion would greatly improve this manuscript and make it more impactful. In addition, the figures are poorly done. They are too small and too contrasted. The live fluorescence images appear overly adjusted to remove background. The EM data are also too small. More 3-D moves of the tomograms would better represent the data. I don't see legends for the movies. Movies corresponding to Figure 3 would be helpful, as these data are central to the model and conclusions. This paper has potential, and I wish I could see more of the data.

We thank reviewer 1 for these comments. The paper is now presented with more exhaustive explanations, with appropriate subheadings and sections, figures have been enlarged, movies have been also included, suggested experiments have been performed as detailed below in the point-by-point responses. Regarding live cell imaging, we have uploaded the movie at low contrast and high contrast, shown in parallel. Given the low level of signal of individual ONM-derived vesicles captured by endolysosomes, we had to increase the contrast to show it.

1. In Fig 2C, SUN1 is looked at. SUN2 would be the more canonical SUN. SUN2 needs to be examined.

SFigs. 2B, 2C now show that endogenous SUN2 is not delivered to the endolysosomal compartments at steady state and during recovery from ER stress.

2. The tomograms in 1D and SF1A are impressive, and I'm happy the n is large. However, I feel like there is a missed quantification opportunity. The data in SF1 should be averaged somehow. Also, the way the blue boxes are drawn is confusing. The model at the top of SF1 is a model and should not be presented as fact. Remove the cartoon from Sup Fig 1.

The new Fig. 1 and SFig 1 now show violin plots that highlight ONM:INM distances as examined by RT-TEM (5 MEF for each experimental condition, Fig. 1L), and probability that the ONMN:INM distance exceeds the 50 nm (SFig. 1D). Moreover, as suggested by the referee, we have removed the model from SFig. 1 and have replaced the blue boxes with a line showing the position of the 50 nm distance in the tomograms.

3. The data in Fig 1 are key to the paper. However, NESPRIN3 is overexpressed from a CMV promoter in pCDNA3.1. It is highly likely that the vast overexpression of NESPRIN3-HALO is not

interacting with a SUN protein. If there is more NESPRIN3 than SUN, than it could more easily be pulled away from the ONM.

4. Related to the above point, look at endogenous NESPRIN3 and 1 and/or 2 also. Seeing multiple endogenous ONM components moving toward the lysosomes would support your model more strongly.

5. FRAP experiments could help determine the dynamics of NESPRIN3. This could be confirmed by mutating the Cys residues, knocking down SUN, and knocking down TMX4. Overexpressed NESPRIN3 is likely to be more dynamic.

6. Which isoform of NESPRIN3 is examined? The isoform that interacts with plectin or not? This could complicate analyses.

In this work, we use ectopically expressed HALO-NESPRIN3 α (Figs. 2B, 3, 5, 10A-10D, S Figs. 2A, 2D, 3C) and endogenous (Figs. 4A-4D, 5G, 5H, 10A-10D) and ectopically expressed (Figs. 5E, 5F) SEC62 as markers of the ONM. The commercially available antibodies that we used to reveal endogenous NESPRINS in imaging analyses (Nesprin 1 Antibody catalog ID: MA5-18077, Invitrogen; Nesprin 2 Antibody catalog ID: LS-C82751 / 223992, LifeSpan BioSciences; Nesprin 3 Antibody catalog ID: ab123021, Abcam) weakly stained the NE and/or showed cross-reactions in our cells. This did not allow monitoring the fate of endogenous NESPRIN proteins in CLSM. Please note that Figs. 8B, 8C show that endogenous NESPRIN proteins (and not SUN proteins) are clients of the TMX4 reductase, which is confirmed with recombinant proteins in Fig. 9.

The morphometric analyses by RT-TEM and FIB-CET prove that the ONM is deformed (in cells exposed to CPA, Figs. 1E, 1F, 4B-4E, 6C, 6D) and vesiculates (during the recovery phase, Figs. 1H, lower panel, 2A, 4C). These micrographs are from non-transfected cells expressing endogenous levels of NESPRIN and SUN proteins and prove that the pathways described in our work are not triggered by NESPRINs overexpression. Moreover, experimental conditions that inhibit deformation of the ONM (e.g., the knockdown (Figs. 10A-10C) or knockout of TMX4 as shown in the new Figs. 10E, 10F) also prevent delivery of recombinant NESPRIN to the endolysosomal compartment, which we use as ONM marker in our study. The knockout of TMX4, where the NE is not expanding when cells are exposed to ER-stress supports our model of remodeling of the NE upon ONM reductase-driven disassembly of LINC complexes. We thank the referee for the insightful suggestion about FRAP experiments, which are part of an external collaboration and will not be included in this submission.

7. In the models here, TMX4 is predicted to be specific to the ONM. But in Cheng et al 2019 (ref 34), digitonin experiments suggest TMX4 is enriched at the INM. If it's at the INM, how could it reach the SUN-KASH disulfide bond? Immuno-EM like those done for Sec62, should be done for TMX4 before you can claim its enriched at the ONM. If it were at the ONM, how would it get there?

In the previous submission, we remained cautious on the nuclear localization of TMX4. Based on IF data by Cheng et al and our own immunofluorescence data (now in Fig. 7E), we wrote that TMX4 is enriched in the NE. We also considered that both if located at the INM or at the ONM, the active site of TMX4 would face the periplasmic space and would, in principle, get

access to the intermolecular disulfide bond linking SUN and NESPRIN proteins in the LINC complex.

Inspired by the referee's comment, we checked by RT-TEM the localization of endogenous TMX4. Our analysis by immuno-gold EM shows the subcellular distribution of endogenous TMX4 in the ONM and the MEF's ER (Fig. 7G and inset), which corresponds to the intracellular distribution of endogenous SEC62 (Figs. 4A-4D). As a comparison, the distribution of endogenous TEX264 that locates in the ER, and both in the ONM and in the INM (Fig. 7H).

8. In Fig 4D, a siRNA resistant construct should be used to rescue the TMX4 phenotype.

Data on silencing and KO of TMX4 are now shown in the new Fig. 10.

In mock-treated cells, both SEC62 and NESPRIN3 are delivered to the endolysosomal compartment for clearance (Figs. 10A, 10C). We remind that our studies show that SEC62 is the membrane-bound receptor for recov-ER-phagy (Fumagalli et al 2016, Loi et al 2019) and for ONM-phagy (this work, Figs. 5C-5F). Upon silencing of TMX4 expression with small interfering RNA (S Figs. 5A-5B show efficiency of transcript and protein down-regulation), we observe the unperturbed lysosomal delivery of SEC62 and the significant inhibition of lysosomal delivery of NESPRIN3 (Figs. 10B, 10C). This shows that TMX4 is dispensable for SEC62-dependent ER-phagy but is required for SEC62-dependent ONM-phagy as explained in the text (page 8, lines 344-369).

Overexpression of TMX4 does not rescue the phenotype (page 8, line 362-366, lines 362-369). Rather, TMX4 overexpression in wild type cells, which does not affect SEC62-dependent ER-phagy (Fig. 10D, Inset 1), substantially inhibits SEC62-dependent ONM-phagy (Fig. 10D, Inset 2). Thus, the phenotype of TMX4 silencing and of TMX4 overexpression is the same (i.e., irrelevant on recov-ER-phagy and strongly inhibitory on autophagy of the ONM).

To better understand the role of TMX4 in NE dynamics, we monitored responses to ER stress in MEF, where TMX4 has been deleted upon CRISPR/Cas9 genome editing (S Figs. 5C, 5E show the control of the KO for transcript and protein). Deletion of TMX4 is harmful to cells, which must be examined at low passage number. Nevertheless, CRISPR-TMX4 cells respond to exposure to low doses of CPA by up-regulating BiP transcript (S Fig. 5D) and protein (S Fig. 5E) shown as conventional ER stress markers. However, consistent with a role of TMX4 in NE dynamics, the NE of CRISPR-TMX4 cells does not expand during CPA-induced stress (Figs. 10E vs 10F).

9. Intermediate disulfide bonds are seen between TMX4 and NESPRINS by multiple assays. But no TMX4-SUN disulfide bonds were looked for in the gels in 4B, and they were not seen in the mass spec. Why are there no SUN-TMX4 intermediates? Also, why wasn't NESPRIN3 pulled down in the mass spec? Can you do the opposite, IP with NESPRIN3 and see if you specifically pull down TMX4?

Resolution of an *inter*-molecular disulfide bonds occurs upon nucleophilic attack of a reductase catalytic cysteine to a cysteine residue of one of the partners. Our data imply that TMX4 attacks (i.e., forms mixed disulfides) NESPRIN proteins preferentially.

This part is better explained and is implemented with new data that highlight the client-specificity of TMX3 (an *oxidase* that distributes in the ER and NE, Fig. 7D), TMX4 (a *reductase* that distributes in the ER and NE, Fig. 7E) and TMX5 (a natural trapping redox enzyme that also distributes in the NE, Fig. 7F). We now show the endogenous proteins that associate with all 3 members of the PDI superfamily to highlight their client's specificity (Figs. 8B, 8C). Only TMX4 associates with members of the NESPRIN family (and, interestingly, with other disulfide-containing proteins possibly involved in NE dynamics such as SLC3A2/CD98, which has a reported activity in viral particle budding from the NE (Hiroata et al JVI 2015), which requires an intermediate step of "detachment" of the INM from the ONM (Hagen et al Cell 2015). We do not have an explanation for the lack of NESPRIN3 in our TMX4-interactome analyses. Possibly the fact that (at least at the transcript level) NESPRIN3 is about 4x less abundant than NESPRIN1 and 2 may be an issue. However, our *in vitro* analyses (Fig. 9) confirm that in cells where the trapping mutant version of TMX4 is co-expressed with NESPRIN3 and SUN1, TMX4 engage NESPRIN3 in mixed disulfides, whereas mixed disulfides with SUN1 remain below detection level (even if SUN1 and 2 levels in these cells are 30 and 50x more abundant, see page 7, lines 294-296).

10. Many references are wrong. For example, references 3-5 are cited throughout for LINC complexes regulating the width of the NE. These references only hypothesize about the width of the NE. Crisp et al 2006 and Cain et al 2014 are the only two places I know of that attempt to answer this question, and neither does so satisfactorily. The idea that LINC complexes regulate set and maintain NE width is still a model. References 6 and 7 have nothing to do with NESPRIN or LINC. Ref 21 is a review, instead it should cite a paper showing what the two sentences are claiming about NESPRIN3 and SUN1. I suspect there are many more examples.

We thank the reviewer for this comment that help us improving our manuscript.

11. Antibody verification is required for commercial antibodies against sSe62 and NESPRIN3. Or we need to be pointed to where the antibodies used have been verified to be specific.

Antibody specificity is shown for endogenous SEC62 and FAM134B in SFig. 3, and for TMX4 in SFig. 5, for GFP-SUN1 and HALO-NSP3 in Fig. 9 (lanes 5 vs. 6-8 and 9 vs. 10-12, respectively).

12. The sentence on line 238 is overblown. This is already an active field!

We thank the referee for the insightful comments/suggestions.

Reviewer #2 (Remarks to the Author):

In this manuscript, the authors analyze the morphology of the nuclear envelope when mammalian cells are treated with the ER stress-inducing compound CPA. They discovered that ER stress causes the swelling of perinuclear space and during recovery, portions of ONM undergo autophagic removal in a manner depending on the recov-ER-phagy receptor SEC62.

Interestingly, the authors find that the transmembrane reductase TMX4 specifically forms mixed disulfide bonds with NESPRIN proteins, which are the ONM components of LINC complexes. Knocking down TMX4 reduces the autophagic delivery of NESPRIN3 to endolysosomes. This work identifies a new ER-stress induced cellular process and unveils a new mechanism regulating LINC complexes.

Major points:

1. I think the identification of TMX4 as a reductase disassembling LINC complexes is the most exciting finding of this manuscript. Unfortunately, the loss-of-function phenotype generated using siRNA is rather weak and unconvincing, presumably because the knockdown is incomplete.

Is it possible for the authors to generate TMX4 knockout cell lines? In addition, it will be interesting to analyze whether the loss of TMX4 prevents the increase of the width of the perinuclear space during ER stress.

This is an important point, and we thank the referee to have raised it.

The statistical relevance of the loss-of-function phenotype upon TMX4 silencing is shown in the new Figs. 10A-10C.

Moreover, as requested, we have generated TMX4-KO MEF upon CRISPR/Cas9 genome editing (Figs. 10E-10F for NE ultrastructure at steady state and in cells exposed to CPA, and SFig. 5C-5E for control of the KO for TMX4 transcript and protein). Deletion of TMX4 is harmful to cells, which must be examined at low passage number (lines 374-384).

Nevertheless, CRISPR^{TMX4} cells respond to exposure to low doses of CPA by up-regulating BiP transcript (SFig. 5D) and protein (SFig. 5E). However, consistent with a role of TMX4 in NE dynamics, the NE of CRISPR^{TMX4} cells does not expand during CPA-induced stress (Figs. 10E vs 10F).

Minor points

2. In the manuscript, the authors used both the term "periplasmic space" (e.g., in the abstract) and the term "perinuclear space" (e.g., in Figure 3B legend). I think "perinuclear space" is a better term and does not cause confusion with bacterial periplasmic space. The authors may want to consider using "perinuclear space" throughout the manuscript.

We thank the reviewer for this suggestion. Perinuclear space has been used in the new submission.

3. In the abstract, "Here we report that expansion of the mammalian ER upon homeostatic perturbations is transmitted to the NE, where the ONM forms large bulges." I think "ONM forms large bulges" is a potentially misleading way to describe the morphological changes. It seems to me that the increase of the width of the perinuclear space is a more prominent

phenomenon than the formation of ONM bulges. The authors may want to consider changing "the ONM forms large bulges" to "the width of the perinuclear space increases".

We modified as suggested, in the abstract and throughout the text.

4. In Fig. 3G, why are mixed disulfide bonds only drawn between the reductase and NESPRIN proteins but not between the reductase and SUN proteins?

The schematics has been removed and replaced by Fig. 6G.

However, we note that the drawing of mixed disulfides between TMX4 and NESPRIN proteins and not between TMX4 and SUN proteins would be justified based on results in Figs. 8 and 9, which show the preference for the trapping mutant of TMX4 to engage endogenous NESPRIN proteins in mixed disulfides. *In cellula* data have now been more thoroughly confirmed with ectopically expressed proteins (Fig. 9) (please also refer to referee 1, response 9).

5. Page 2, line 49. "Analyses of 17 tomograms reveals that at steady state". "reveals" should be "reveal".

This has been corrected, thank you.

Reviewer #3 (Remarks to the Author):

In the manuscript by Kucińska, Fedry et al., the authors describe an ER-phagy pathway specific for the clearance of portions of the outer nuclear membrane. They ultrastructurally describe changes in ER morphology in different cell culture systems upon inhibiting the sarco/ER calcium pump. By various EM methods including FIB/SEM cryo-ET, they show that after treatment and during recovery the distance between the inner and outer nuclear membrane increases. At these sites Nesprin3 dissociates from the LINC complex and is found in LAMP1-positive endolysosomes as shown by fluorescence microscopy experiments. They link this degradation to the selective autophagy receptor Sec62, since deletion of this protein inhibits Nesprin3 targeting to endolysosomes. Finally, their experiments suggest that the dissociation of Nesprin3 from the LINC complex is mediated by the oxidoreductase TMX4. Knockdown of TMX4 indeed blocks Nesprin3 accumulation in endolysosomes but does not affect Sec62 accumulation.

The identification of a ERphagy pathway specific for the degradation of the outer nuclear membrane is an exciting finding and the manuscript is well written. However, additional experiments are needed to strengthen the conclusion that a Sec62 dependent macroautophagy pathway is responsible for outer nuclear membrane degradation.

Major points:

1. In order to establish Sec62 as the receptor for this selective autophagy pathway the authors

should replace Sec62 by the LIR mutant to separate the translocon function of Sec62 from its involvement in autophagy.

The experiment suggested by the referee is shown in the new Fig. 5, where SEC62 (Fig. 5E, quantification in Fig. 5D) or the LIR mutant of SEC62 (Fig. 5F, quantification in Fig. 5D) have been back-transfected in the SEC62-KO background (shown in Figs. 5C, 5D). Only the wild type form of SEC62 restores lysosomal delivery of the ONM (also refer to Fig. 5D for the quantification and SFig. 3A for controls of the knockout and of the back-transfections).

2. It is necessary to further strengthen the point of a novel macroautophagy pathway by characterizing it in more detail. Does this pathway lead to Sec62 dependent Nesprin3 degradation? Given that Atg8ylation is a sensor for membrane stress and not necessarily involved in macroautophagy the authors should test other autophagy mutants which are not involved in Atg8 lipidation.

3. CLEM experiments showing Nesprin3 positive vesicles within autophagosomes would be another strong indication of the existence of an ONM specific autophagy pathway.

NESPRIN3-positive ONM portions are degraded within RAB7/LAMP1-positive endolysosomes, where it accumulates only upon inhibition of the hydrolytic activity (SFig. 2A for an experiment without BafA1). During recovery from ER stress, RAB7/LAMP1-positive endolysosomes directly capture NESPRIN3-positive portions of the ONM for clearance (Figs. 2B, 3A, 3B, 3E-3H, 4E, 5, 10, Movie 3). These data are consistent with a process corresponding to the piecemeal *micro*-autophagy that we previously reported to control LC3-dependent lysosomal clearance of excess ER during recovery from ER stress (recov-ER-phagy, references Fumagalli et al 2016 and Loi et al 2019 are given in the text). Both recov-ER-phagy and ONM-phagy rely on direct engulfment of organelle-derived vesicles by LAMP1/RAB7-positive endolysosomes (for ONM-phagy this is shown in Fig. 2B and Movie 3), on autophagy genes that regulate LC3 lipidation (Figs. 3E-3F), but not on ATG14 (Figs. 3G-3H), which is dispensable for LC3 lipidation but is involved in autophagosome formation and in control of membrane tethering and autophagosome fusion with lysosomes (references in the manuscript, page 4, lines 178-181). Both recov-ER-phagy and ONM-phagy involve the autophagy receptor SEC62 (Fig. 5, SFig. 3A) but not the starvation-induced ER-phagy receptor FAM134B (S Figs. 3B-3D). The significant difference between the two pathways induced during cell recovery from ER stress is that ONM-phagy relies on TMX4 intervention (to disassemble LINC complexes, Figs. 10A-10C), whereas recov-ER-phagy does not involve TMX4 activity (Figs. 10A-10C).

4. Do autophagy components such as LC3 colocalize with Nesprin3 at the nuclear envelope during vesicle formation?

We do see the autophagy receptor SEC62 (both endogenous and ectopically expressed) co-localizing with NESPRIN3 in the NE (Figs. 5E-5H) in immunofluorescence (CLSM). This is confirmed by immunogold-TEM, which reveals endogenous SEC62 in the ONM both at steady state (Fig. 4A) and during recovery from ER stress (Figs. 4B-4D). SEC62 also localizes at the limiting membrane of ONM protrusions (Fig. 4B), of ONM-derived vesicles in proximity of

degradative endolysosomes (Fig. 4C), or at contact sites between the ONM and endolysosomes (Fig. 4D). Endogenous LC3 only sporadically co-localizes with NESPRIN3 in the NE, but certainly do so within the LAMP1-positive compartment (SFig. 2E).

5. From many of the cryo-ET tomograms it is not clear that the vesicles described to pinch off the ONM are next to autophagic structures or endolysosomes as shown by fluorescence microscopy. Only Figure 2M shows this. Are these described vesicles really the target of autophagy, since potentially other pathways such as COPI/II could act on the nuclear envelope (PMID: 8548805). For this the authors should perform CLEM experiments using Nesprin3 to guide tomogram acquisition.

Fluorescence microscopy, in particular life cell imaging (Fig. 2B and Movie 3) proves that NESPRIN3-positive ONM portions that pinch off during the recovery from ER stress are captured (directly) by endolysosomes. Despite the rapidity of the events, several RT-TEM micrographs taken during the recovery phase from ER stress show endolysosomes in proximity of the ONM. In the new version of the manuscript, we selected three images. In the first, an ONM-derived vesicle, which is detaching from the ONM and displays the autophagy receptor SEC62 at the limiting membrane (immunogold labeling) is shown next to an endolysosome (Fig. 4C and inset, arrow 1). In the second (Fig. 4D and Inset), a site of contact between the ONM and an endolysosome is shown (the arrow indicates immunogold labeled SEC62). In the third, an endolysosome containing NESPRIN3-positive membranes (Fig. 4E, arrowheads) is next to the NE displaying NESPRIN3 at the ONM (Fig. 4E, arrow).

Additional points:

6. The images in Figure 3A do not allow to see the densities the authors have segmented in Figure 3B. Movies of the tomograms should be provided. In addition, it is unclear how many tomograms were analyzed? It would be good to do the segmentation (if possible) in an automated fashion as for example described in this publication (PMID: 31907446).

We thank the reviewer for this comment. We now better explain this in the manuscript. The Figure 3A (first submission) is now panels A-F in Fig. 6. For the segmentation, we chose the best tomogram in each condition as an illustration. Indeed, visualizing thin filaments in the perinuclear space is only possible when lamellae are extremely thin in this region (most lamellae prepared at the perinuclear region of mammalian cells, typically tend to be quite thick). We now clarified the manuscript on this point (pages 5, 6, lines 227-236): “To visualize the PNS in vitrified MEF cells, we selected the thinnest FIB-CET tomogram in each condition, giving the highest contrast in the NE region. We deconvoluted the tomogram and masked stronger densities to guide the manual segmentation of densities connecting the ONM and INM. The analyses of cells at steady state (Figs. 6A, 6B, S Figs. 4A, 4B, Movie 4.1), upon exposure to ER stress (Figs. 6C, 6D, S Figs. 4C, 4D, Movie 4.2), or during recovery from ER stress (Figs. 6E, 6F, S Figs. 4E, 4F, Movie 4.3) indicates the presence of continuous densities bridging the INM and ONM only in NE subdomains, where the distance between the lipid bilayers is below the 50 nm (Figs. 6B, 6D, 6F and arrowheads in S Fig. 4). Above this

limit, we did not observe filaments connecting the INM and the ONM (Figs. 6B, 6D, 6F, S Figs. 4A-4F, Movies 4.1-4.3).” **Automated segmentation as referred to by the reviewer is possible for large strong features like membrane, although in practice it is typically refined manually before applying quantification methods (ref. Barad et al. in the text). We did not apply automatic methods for our small filaments, but we now clarify in the methods section that a density threshold mask was first applied to the tomogram to highlight only stronger densities and guide subsequent manual tracing steps (legend Fig. 6, S Fig. 4 and Methods section, page, 29, lines 884-888). In addition, we provide Movie4.1, Movie4.2, Movie 4.3) for the tomograms and S Fig. 4 showing different Z planes for the tomograms, the same planes in the deconvoluted tomograms, and arrow heads pointing to the strong filament-like densities that we segmented.**

7. One would expect that the Nesprin3 signal always colocalize with the Sec62 signal inside of endolysosomes. Is this the case? Would be nice to show a merged image between the Nesprin and Sec62 signal in Figure 4C.

The co-localization between NESPRIN3 and endogenous SEC62 during recovery from ER stress is shown in the new panels in Figs. 5G, 5H.

8. For Mov 1: It would be nice to have the movie played without the segmentation first to access the raw data.

As requested, we have added a second version of the movie (Movie 2A in the new submission), without segmentation.

9. Figure S3A is not very clear. Vesicles in the inlay are only to imagine. The authors should enhance the brightness/contrast of this image and mark the Nesprin3 positive vesicles.

This panel has been removed. NESPRIN3-positive vesicles within LAMP1/RAB7-positive compartments are now shown in Figs. 2, 3, 5, 10 (in CLSM) and in Fig. 4E (RT-TEM).

10. LINE 113 – 116: “In yeast, selective autophagy of nuclear components (i.e., nucleus-derived double membrane vesicles, NE domains containing specific nucleoporins, nuclear pore complexes) has mainly been studied as a catabolic process induced upon nutrient deprivation and it involves the autophagy receptor Atg39.” In the case of NPCs there are additional receptors suggested in the literature (PMID: 32029894; PMID: 32453403)

Thank you, this paragraph has been deleted from the new version of the manuscript.

REVIEWERS' COMMENTS

Reviewer #1 (Remarks to the Author):

This revised manuscript is significantly improved. It represents an important advance in the role of TMX4 at the nuclear envelope to regulate LINC complexes. It is much better written and the experiments are fully controlled.

Reviewer #2 (Remarks to the Author):

The revision has addressed all my concerns and I support the publication of this paper.

Reviewer #3 (Remarks to the Author):

All of my concerns have been addressed, and I congratulate the authors on this well-revised version of the manuscript.